# RGNMR: A Gauss-Newton method for robust matrix completion with theoretical guarantees

**Eilon Vaknin Laufer**
Weizmann Institute of Science
eilon.vaknin@weizmann.ac.il

**Boaz Nadler**
Weizmann Institute of Science
boaz.nadler@weizmann.ac.il

## Abstract

Recovering a low rank matrix from a subset of its entries, some of which may be corrupted, is known as the robust matrix completion (RMC) problem. Existing RMC methods have several limitations: they require a relatively large number of observed entries; they may fail under overparametrization, when their assumed rank is higher than the correct one; and many of them fail to recover even mildly ill-conditioned matrices. In this paper we propose a novel RMC method, denoted `RGNMR`, which overcomes these limitations. `RGNMR` is a simple factorization-based iterative algorithm, which combines a Gauss–Newton linearization with removal of entries suspected to be outliers. On the theoretical front, we prove that under suitable assumptions, `RGNMR` is guaranteed exact recovery of the underlying low rank matrix. Our theoretical results improve upon the best currently known for factorization-based methods. On the empirical front, we show via several simulations the advantages of `RGNMR` over existing RMC methods, and in particular its ability to handle a small number of observed entries, overparameterization of the rank and ill-conditioned matrices. In addition, we propose a novel scheme for estimating the number of corrupted entries. This scheme may be used by other RMC methods that require as input the number of corrupted entries.

## 1 Introduction

Low-rank matrices play a fundamental role in multiple scientific disciplines. As reviewed in Davenport and Romberg (2016), in various applications, there is a need to recover a low rank matrix from only a subset of its entries. Examples include recommendation systems (Bennett et al., 2007), various problems in computer vision (Kennedy et al., 2016; Tulyakov et al., 2016; Miao and Kou, 2021), in sensor networks (Wu et al., 2025) and in single-cell data analysis (Lejun et al., 2025). A key challenge in these and other applications is that some of the observed entries may be arbitrarily corrupted outliers.

In this work, we consider the robust matrix completion (RMC) problem, of recovering a low rank matrix from a subset of its entries, out of which a few are outliers. Formally, let $X \in \mathbb{R}^{n_1 \times n_2}$ be a matrix with a decomposition $X = L^* + S^*$, where $L^*$ is a rank $r$ matrix and $S^*$ is a sparse corruption matrix whose few non-zero entries are arbitrary. The RMC problem is to recover $L^*$ from the subset $\{X_{i,j} \mid (i,j) \in \Omega\}$, where $\Omega \subset [n_1] \times [n_2]$ denotes the subset of observed entries in $X$.

**Related work.** Over the past decade, the RMC problem was studied from both mathematical, computational and statistical perspectives. On the computational front, several RMC algorithms have been proposed. In general, to solve the RMC problem the following optimization issues need to be addressed: (i) either promote a low rank solution or strictly enforce it; and (ii) suppress the potentially detrimental effect of the outliers, whose locations are a-priori unknown.

39th Conference on Neural Information Processing Systems (NeurIPS 2025).

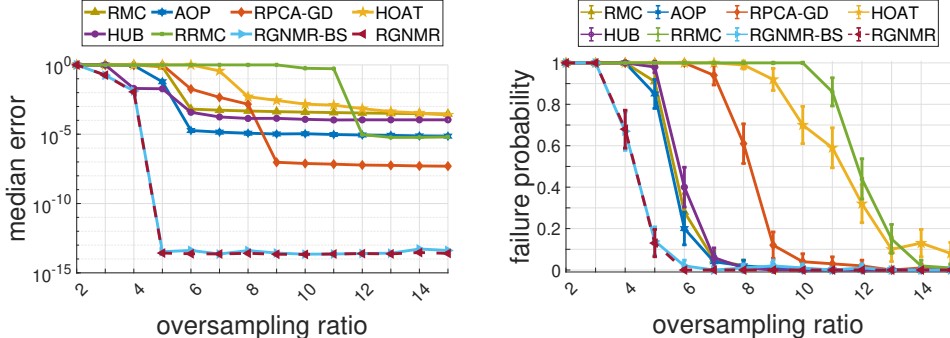

Figure 1: Performance of RMC methods as a function of the number of observed entries measured by the oversampling ratio $\frac{|\Omega|}{r \cdot (n_1 + n_2 - r)}$ where $r \cdot (n_1 + n_2 - r)$ is the number of degrees of freedom of a rank $r$ matrix. (left) Median `rel-RMSE`; (right) Failure probability ($\pm 1.96$ SE). The underlying matrix $L^*$ has a condition number $\kappa = 2$. The corruption fraction is $\alpha = 5\%$.

Regarding the first issue, one approach is to incorporate into the optimization objective a low-rank promoting penalty such as the nuclear norm (Candès et al., 2011; Nie et al., 2012; Klopp et al., 2017; Wong and Lee, 2017; Chen et al., 2021). Under suitable assumptions on the matrices $L^*$, $S^*$ and on the set $\Omega$, these methods enjoy strong theoretical guarantees. However, these schemes are in general computationally slow, and do not scale well to large matrices. A different approach is to strictly enforce a rank $r$ solution. This can be done by optimization over the manifold of rank $r$ matrices (Yan et al., 2013; Cambier and Absil, 2016), or by projecting matrices onto it. Several works factorize the target matrix as $L = UV^\top$, and optimize over the two factor matrices $U \in \mathbb{R}^{n_1 \times r}$ and $V \in \mathbb{R}^{n_2 \times r}$ (Yi et al., 2016; Lin et al., 2017; Zeng and So, 2017; Huang et al., 2021; Wang et al., 2023). The resulting problem involves only $(n_1 + n_2)r$ variables instead of the $n_1 \times n_2$ entries in $L$. Factorization based methods are thus in general more scalable and able to handle larger matrices.

Regarding the second issue, some works attenuate the effect of the outliers either by estimating the locations of the corrupted entries and removing them, or by estimating their values $S_{ij}$ for $(i, j) \in \Omega$ (Yan et al., 2013; Cherapanamjeri et al., 2017; Huang et al., 2021; Chen et al., 2021; Wang and Wei, 2024). Other works mitigate the influence of outliers by using a robust norm on the difference $X - L$ in the objective function (Nie et al., 2012; Cambier and Absil, 2016; Lin et al., 2017; Zhao et al., 2016; Zeng and So, 2017; Wang et al., 2023).

**Limitations of existing RMC methods.** Despite extensive research on the RMC problem, current algorithms have various limitations: (i) Several RMC methods fail to recover the low rank matrix unless the number of observed entries is quite large; (ii) Some methods require as input the (often unknown) rank $r$ of the target matrix and fail when overparameterized even by just a single additional dimension (an input rank of $r + 1$); (iii) Various methods fail to recover a matrix with a moderate condition number, as low as 5.

We illustrate these issues in Figures 1, 2 and 3. These figures show recovery results of various RMC algorithms for a matrix $L^*$ of size $3200 \times 400$, rank $r = 5$, at a corruption rate of $\alpha = 5\%$ (see Section 4 for further details). As in Zeng and So (2017); Tong et al. (2021) and Huang et al. (2021), the quality of a recovered matrix $\hat{L}$ is measured by the relative error `rel-RMSE` $= \frac{\|\hat{L} - L^*\|_F}{\|L^*\|_F}$, and is considered a failure if `rel-RMSE` $> 10^{-3}$. For each simulation setting we ran 100 independent realizations. As shown in these figures, RMC methods such as `AOP` (Yan et al., 2013), `RMC` (Cambier and Absil, 2016), `RPCA-GD` (Yi et al., 2016), `RRMC` (Cherapanamjeri et al., 2017), `HUB` (Ruppel et al., 2020) and `HOAT` (Wang et al., 2023) require a large number of observed entries and fail when overparameterized. In addition, all of them (except `HUB`) fail at condition numbers greater than 5.

On the theoretical front, some RMC methods have no theoretical recovery guarantees. The currently available guarantees for other methods are somewhat limited. As detailed in Table 1, the guarantees for some methods only hold for fully observed matrices, or their allowed corruption level strongly depends on the matrix rank and condition number.

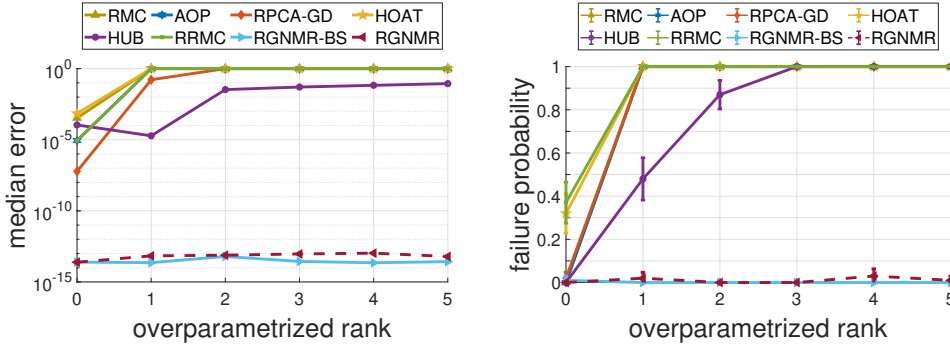

Figure 2: Performance of RMC methods under overparameterization with input rank of $5 + i$ for $i \in [0, 5]$. (left) Median `rel-RMSE`; (right) Failure probability ($\pm 1.96$ SE). The matrix $L^*$ has a condition number $\kappa = 2$ and the oversampling ratio is $\frac{|\Omega|}{r \cdot (n_1 + n_2 - r)} = 12$. The corruption fraction is $\alpha = 5\%$

These shortcomings raise the following challenges: (i) *Develop a computationally efficient RMC method able to handle ill-conditioned matrices, overparameterization, and a small number of observed entries; (ii) derive for this method strong recovery guarantees that improve upon those of existing methods.*

**Our contributions.** In this work we make a step towards resolving this challenge. We propose a novel RMC method, denoted `RGNMR`, which overcomes the above limitations. Our proposed method is a simple iterative algorithm based on Gauss–Newton linearization and removal of entries suspected to be outliers. As seen in Figures 1, 2 and 3, empirically `RGNMR` can successfully recover the underlying matrices with significantly fewer observed entries than existing methods. Furthermore, the performance of `RGNMR` is not affected by an overparameterized rank or by the condition number. In Appendix B we show that `RGNMR` continues to outperform other methods under broader conditions, including a combination of outliers and additive noise, non-uniform sampling, higher rank matrices, a range of fractions of outliers, etc. Furthermore, we show that `RGNMR` performs well on a real data set from computer vision, involving background extraction. In addition to its excellent empirical performance, in Section 3 we derive theoretical recovery guarantees for `RGNMR`. Specifically, we present two theorems. Theorem 3.1 states that given a suitable initialization, `RGNMR` recovers the target matrix at a linear rate. This theorem holds under the weakest known assumptions for factorization based RMC methods. Theorem 3.2 establishes that the initialization scheme we propose yields a sufficiently accurate initial estimate for Theorem 3.1 to hold.

## 2 The RGNMR algorithm

To describe `RGNMR`, we first introduce some notation. For a subset $\Omega \subseteq [n_1] \times [n_2]$, we denote by $\mathcal{P}_\Omega$ the following projection operator

$$\mathcal{P}_\Omega(A)_{i,j} = \begin{cases} A_{i,j} & (i,j) \in \Omega, \\ 0 & (i,j) \notin \Omega. \end{cases}$$

For a matrix $A$ and a set $\Omega$, we denote $\|A\|_{F(\Omega)}^2 = \|\mathcal{P}_\Omega(A)\|_F^2 = \sum_{(i,j) \in \Omega} A_{i,j}^2$. Finally, we denote the set of corrupted entries by $\Lambda_* = \{(i,j) \in \Omega \mid S_{i,j}^* \neq 0\}$ and the number of corrupted entries by $k^* = |\Lambda_*|$. Both $\Lambda_*$ and $k^*$ are unknown.

The optimization variables of `RGNMR` are two factor matrices $U \in \mathbb{R}^{n_1 \times r}, V \in \mathbb{R}^{n_2 \times r}$ and a subset $\Lambda \subset \Omega$ which estimates the locations of the corrupted entries. `RGNMR` receives as input the following quantities; the subset of observed entries $\{X_{i,j} \mid (i,j) \in \Omega\}$; the rank of the target matrix $r$; an upper bound $k$ on the number of corrupted entries; initial guess $(U_0, V_0)$ for the factor matrices and $\Lambda_0 \subset \Omega$ for the set of corrupted entries.

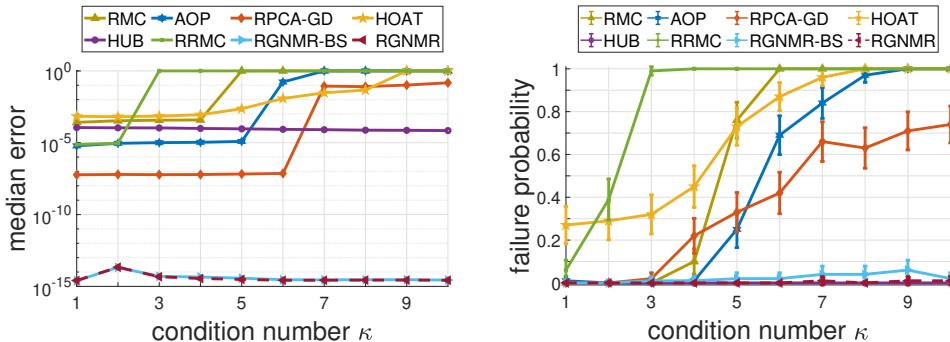

Figure 3: Performance of RMC methods as a function of the condition number. (left) Median `rel-RMSE`; (right) Failure probability ($\pm 1.96$ SE). The oversampling ratio is $\frac{|\Omega|}{r \cdot (n_1 + n_2 - r)} = 12$. The corruption fraction is $\alpha = 5\%$.

At each iteration `RGNMR` performs the following two steps: (i) given the current set of suspected outlier entries $\Lambda$, update $U, V$ using the remaining entries $\mathcal{P}_{\Omega \setminus \Lambda}(X)$; (ii) given the updated matrix $L$, recompute the new set of suspected outliers $\Lambda$, by the $k$ entries with largest magnitude in $\mathcal{P}_\Omega(X - L)$.

Let us provide some motivation for the above two steps. If the estimated set of non corrupted entries, $\Omega \setminus \Lambda$, is large enough and contains *only* non-corrupted entries, namely $\Lambda_* \subset \Lambda$, then the matrix $L^*$ could be recovered exactly from $\mathcal{P}_{\Omega \setminus \Lambda}(X) = \mathcal{P}_{\Omega \setminus \Lambda}(L^*)$ by solving a vanilla matrix completion problem, with no outliers or noise. Regarding the second step, consider an ideal case where after step (i) of some iteration `RGNMR` obtained $L = L^*$. Then at all the non-corrupted entries $(i, j) \in \Omega \setminus \Lambda_*$, the residual $X_{i,j} - L_{i,j} = 0$ and $L^*$ is a fixed point of the algorithm. Therefore, given $L$ it is reasonable to estimate the corrupted entries by the entries with largest magnitude in $\mathcal{P}_\Omega(X - L)$. Iterating this process we hope to identify a sufficiently large subset of $\Omega \setminus \Lambda_*$ and consequently recover $L^*$.

We now describe the two steps of `RGNMR`. The first step builds upon the matrix completion approach of Zilber and Nadler (2022). Formally, given the estimate $\Lambda_t$ of $\Lambda_*$ and the current factorization $(U_t, V_t)$, the updated matrices $(U_{t+1} V_{t+1})$ are the solution of the following least squares problem,

$$(U_{t+1}, V_{t+1}) = \arg\min_{U,V} \mathcal{L}^t_{\Omega \setminus \Lambda_t}(U, V) = \arg\min_{U,V} \|U_t V^\top + U V_t^\top - U_t V_t^\top - X\|_{F(\Omega \setminus \Lambda_t)}. \quad (1)$$

This problem is rank deficient with an infinite number of solutions. As in Zilber and Nadler (2022), we choose the new estimate to be the solution with minimal norm $\|U_{t+1}\|_F^2 + \|V_{t+1}\|_F^2$.

For the second step, we construct a new estimate of $\Lambda_*$ by the $k$ largest residual entries in (1),

$$\Lambda_{t+1} = \arg\min_{\Lambda \subset \Omega, |\Lambda| = k} \|U_t V_{t+1}^\top + U_{t+1} V_t^\top - U_t V_t^\top - X\|^2_{F(\Omega \setminus \Lambda)}. \quad (2)$$

`RGNMR` iterates these two steps until convergence or until a maximal number of iterations $T$ is reached. Algorithm 1 presents an outline of `RGNMR`.

**Remark 2.1.** *In general, removing some of the observed entries, as in (2), may be detrimental. For example, if after removal of entries suspected as outliers, only less than $r$ entries remain at a specific row or column, then the resulting matrix completion problem is ill-posed and exact recovery of $L^*$ is not possible. As illustrated empirically in Zilber and Nadler (2022), `GNMR` is able to complete low rank matrices even if the number of observed entries is near the information limit. Therefore, as long as the number of remaining observed entries in each row and column is above $r$, our method is often still able to recover the target matrix.*

**Remark 2.2.** *As shown in Zilber and Nadler (2022), in the absence of outliers, `GNMR` which iteratively solves (1), can successfully recover ill-conditioned low rank matrices, from relatively few observed entries. However, as illustrated in Figure 5 in the appendix, `GNMR` is not robust and fails completely in the presence of even a small fraction of corrupted entries. Since in the first few iterations the set $\Omega \setminus \Lambda_t$ typically includes outliers, employing `GNMR` would not result in a significantly better update of $(U_t, V_t)$ than that of a single optimization step. Therefore, in our scheme we do not run `GNMR` till*

**Algorithm 1** RGNMR

**Input:**

- $\{X_{i,j} \mid (i,j) \in \Omega\}$ - observed entries
- $r$ - rank of $L^*$
- $k$ - upper bound on the number of corrupted entries
- $\begin{pmatrix} U_0 \\ V_0 \end{pmatrix} \in \mathbb{R}^{(n_1+n_2)\times r}$ - initialization
- $\Lambda_0$ - initial estimate of the set of corrupted entries
- $T$ - maximal number of iterations

**Output:** $\hat{L}$ of rank $r$

    **for** $t = 0 \dots T-1$ **do**

$$\begin{pmatrix} U_{t+1} \\ V_{t+1} \end{pmatrix} = \arg\min_{U,V} \|U_t V^\top + U V_t^\top - U_t V_t^\top - X\|^2_{F(\Omega \backslash \Lambda_t)}$$

$$\Lambda_{t+1} = \arg\min_{\Lambda \subset \Omega, |\Lambda|=k} \|U_t V_{t+1}^\top + U_{t+1} V_t^\top - U_t V_t^\top - X\|^2_{F(\Omega \backslash \Lambda)}$$

    **end for**
    **return** $P_r(U_{T-1}V_T^\top + U_T V_{T-1}^\top - U_{T-1}V_{T-1}^\top)$

---

*convergence. Instead we re-estimate $\Lambda$ after each update of $(U_t, V_t)$. This approach avoids redundant iterations of GNMR and yields considerable savings in terms of runtime.*

## 2.1 Estimating the number of corrupted entries

One of the input parameters to RGNMR is $k$. At each iteration RGNMR removes the $k$ entries with largest residual error, and performs low-rank matrix completion on the remaining entries. For RGNMR to succeed, it is crucial for $k$ to be a tight upper bound of the true number of corrupted entries $k^*$. Indeed, if $k \gg k^*$ then too many entries are removed and on the remaining entries the corresponding matrix completion problem is ill posed. As the true number of corrupted entries $k^*$ is unknown, it is important to develop a method to tightly upper bound it. Here we propose a scheme to do so.

Our approach is motivated by the empirical observation that the estimates $\Lambda_t$ of RGNMR behave differently if $k \leq k^*$ or $k > k^*$. If $k \leq k^*$ then the estimates $\Lambda_t$ of the set of corrupted entries converge after several iterations. However, if $k > k^*$, the sets $\Lambda_t$ do not converge. This is illustrated in Figure 4 in the appendix, and we provide some intuition for this behavior below. Building upon this observation, we propose a binary search algorithm to estimate $k^*$, assuming it belongs to the range $[k_{\min}, k_{\max}]$. In practice, we may take $k_{\min} = 0$ and $k_{\max} = |\Omega|/2$ (namely at most 50% corrupted entries). At each step we run RGNMR with $k = \lfloor (k_{\min} + k_{\max})/2 \rfloor$. If the sets $\Lambda_t$ converged, then we update $k_{\min} = k$. Otherwise, we set $k_{\max} = k$. After $\log_2(k_{\max} - k_{\min})$ steps we obtain a tight upper bound on $k^*$.

To provide intuition to why RGNMR behaves differently with $k \leq k^*$ or $k > k^*$, consider the following scenario. Assume that $k \leq k^*$ and that at some iteration, the set $\Lambda_t$ of $k$ entries with largest current residuals, satisfies that $\Lambda_t \subseteq \Lambda_*$. The next estimate $L_{t+1}$ of $L^*$ minimizes $\|\mathcal{P}_{\Omega \backslash \Lambda_t}(L - X)\|_F$, ignoring the entries in $\Lambda_t$, which by assumption are all outliers. Hence, $L_{t+1}$ tries to fit a low rank matrix to the remaining entries, and thus we expect the entries in $\Lambda_t$ to remain the $k$ largest residual entries. Therefore, $\Lambda_t = \Lambda_{t+1}$ and as the same argument holds for all subsequent iterations the sets $\Lambda_t$ converge. In contrast, if $k > k^*$ then at any iteration $t$, the set $\Lambda_t$ contains at least $k - k^*$ non-corrupted entries. Assume that at some iteration $t$, the method successfully detected all outliers, i.e. $\Lambda_* \subset \Lambda_t$. Then, if $k$ is not too large, the next estimate should be very close to the true low rank matrix $L_{t+1} \approx L^*$. As the computations are done in finite precision, the residuals at both $\Omega \backslash \Lambda_t$ as well as at $\Lambda_t \backslash \Lambda_*$ are not precisely zero but rather, due to rounding errors, appear as very small random values. Therefore, even though $\Lambda_* \subset \Lambda_{t+1}$ its remaining $k - k^*$ entries are not the same as those of $\Lambda_t \backslash \Lambda_*$, but rather change at each iteration $t$. Hence, the sets $\Lambda_t$ do not converge.

**Algorithm 2** `RGNMR` - modified

**Input:** All the inputs of `RGNMR` and the following additional parameters:

- $\mu$ - incoherence parameter of $L^*$
- $\alpha$ - a bound on the fraction of outliers in each row/column
- $\gamma$ - over-removal factor
- $\delta$ - neighborhood parameter

**Output:** $L^*$ of rank $r$

    **for** $t = 0 \dots T - 1$ **do**

$$\Lambda_t = \text{support}\left(\mathcal{T}_{\gamma\alpha}\left(U_t V_t^\top - X, \Omega\right)\right)$$

$$\begin{pmatrix} U_{t+1} \\ V_{t+1} \end{pmatrix} = \arg\min_{U,V} \left\{ \|U_t V^\top + U V_t^\top - U_t V_t^\top - X\|_{F(\Omega \setminus \Lambda_t)}^2 \,\Big|\, (U,V) \in \mathcal{B}_\mu \cap \mathcal{C}\left(U_t, V_t, \tfrac{\delta}{4^{t+1}}\right) \right\}$$

    **end for**

    **return** $L_T = U_T V_T^\top$

To demonstrate that our binary search scheme successfully bounds the number of corrupted entries we compare two variants of `RGNMR`. The first, denoted simply `RGNMR`, is given the true number of corrupted entries $k^*$ as input, $k = k^*$. The second variant, denoted `RGNMR-BS`, first upper bounds $k^*$ using our binary search scheme and then uses this upper bound $\hat{k}$ as input, $k = \hat{k}$. As illustrated in Figures 1, 2 and 3, `RGNMR-BS` performance is similar to that of `RGNMR`, which implies that our method obtains a sufficiently tight upper bound on $k^*$.

**Remark 2.3.** *Several RMC methods aim to remove the corrupted entries and require an estimate of $k^*$ (Yan et al., 2013; Yi et al., 2016). Since the intuition provided above holds also for these methods, a similar scheme to estimate $k^*$ can be incorporated there as well. We note that in the simulations we conducted these methods were provided with the exact number of outliers. Still, in various settings, these methods failed to recover the underlying matrix, whereas `RGNMR` succeeded. This highlights that our novel scheme for estimating the number of outliers is not the sole advantage of our proposed method.*

## 3 Recovery guarantees

In this section we present recovery guarantees for `RGNMR`. Recall that the goal is to recover a rank $r$ matrix $L^* \in \mathbb{R}^{n_1 \times n_2}$ from a subset $\Omega$ of the entries of $X = L^* + S^*$. We start with some notations and the assumptions made in our theoretical analysis.

**Notations.** The $i$-th largest singular value of the matrix $L^* \in \mathbb{R}^{n_1 \times n_2}$ is denoted by $\sigma_i^*$, and its condition number is denoted by $\kappa = \sigma_1^*/\sigma_r^*$. We denote the operator norm of a matrix (a.k.a. spectral norm) by $\|A\|_{op}$, its Frobenius norm by $\|A\|_F$, its $i$-th row by $A_{(i,\cdot)}$, its $j$ column by $A_{(\cdot,j)}$ its largest row norm by $\|A\|_{2,\infty} \equiv \max_i \|A_{(i,\cdot)}\|$ and its zero norm by $\|A\|_0 = |\{(i,j) \in [n_1] \times [n_2] \mid A_{i,j} \neq 0\}|$. We define the matrix $|A|$ by $|A|_{(i,j)} = |A_{(i,j)}|$. We denote by $P_r(A)$ the best rank $r$ approximation of $A$ in Frobenius norm, namely the projection to the subspace spanned by the $r$ singular vectors with largest singular values. We denote $\Omega_{(\cdot,j)} = \{i \mid (i,j) \in \Omega\}$, $\Omega_{(i,\cdot)} = \{j \mid (i,j) \in \Omega\}$ and $c_j = |\Omega_{(\cdot,j)}|$, $r_i = |\Omega_{(i,\cdot)}|$. The Procrustes distance between $Z_1, Z_2 \in \mathbb{R}^{n \times r}$ is

$$d_P(Z_1, Z_2) = \min\left\{ \|Z_1 - Z_2 P\|_F \mid P \in \mathbb{R}^{r \times r} \text{ is orthogonal} \right\}. \tag{3}$$

For a vector $v$ we denote by $v^{(k)}$ its $k$-th largest entry in absolute value.

**Assumptions.** For the RMC problem to be well-posed, we make three standard assumptions.

**Assumption 1.** *The underlying matrix $L^*$ is incoherent with incoherence parameter $\mu$. Formally, if $U\Sigma V^\top$ is the SVD of $L^*$ then*

$$\|U\|_{2,\infty} \leq \sqrt{\mu r/n_1}, \quad \|V\|_{2,\infty} \leq \sqrt{\mu r/n_2}. \tag{4}$$

Table 1: Recovery guarantees requirements of various RMC methods, up to multiplicative constant factors. The weakest conditions for each category of methods are in bold.

| Method Category | Method | Sample Complexity $(pn_2 \geq)$ | Corruption Rate $(\alpha \leq)$ |
|---|---|---|---|
| Factorization Based | Zheng and Lafferty (2016) | $\max\{\boldsymbol{\mu r \log n_1, \mu^2 r^2 \kappa^2}\}$ | $\alpha = 0$, No Corruption |
| | Tong et al. (2021) | $p = 1$, Fully Observed | $\frac{1}{r^{\frac{3}{2}}\mu\kappa}$ |
| | Cai et al. (2024) | $p = 1$, Fully Observed | $\frac{1}{r^{\frac{3}{2}}\mu\kappa}$ |
| | Yi et al. (2016) | $\mu^2 r^2 \kappa^4 \log n_1$ | $\frac{1}{r\mu\kappa^2}$ |
| | Our Theorem 3.1 | $\max\{\boldsymbol{\mu r \log n_1, \mu^2 r^2 \kappa^2}\}$ | $\boldsymbol{\frac{1}{r\mu\kappa}}$ |
| Full Matrix | Cherapanamjeri et al. (2017) | $\mu^2 r^2 \log^2(\mu r \sigma_1^*) \log^2 n_1$ | $\frac{1}{r\mu}$ |
| | Wang and Wei (2024) | $\mu^3 r^3 \kappa^4 \log n_1$ | $\frac{1}{r^2\mu^2\kappa^2}$ |

For future use, we denote by $\mathcal{M}(n_1, n_2, r, \mu, \kappa)$ the set of $n_1 \times n_2$ matrices of rank $r$, incoherence parameter $\mu$ and condition number $\kappa$.

**Assumption 2.** *[Bernoulli Model] Each entry in $X$ is independently observed with probability $p$. Hence, the number of observed entries is not fixed, but rather $|\Omega| \sim Bin(n_1 n_2, p)$.*

**Assumption 3.** *In each row and column, the fraction of observed entries which are corrupted is bounded. Formally, for a known $0 < \alpha < 1$, we assume that $S^* \in \mathcal{S}_\alpha^\Omega$, where*

$$\mathcal{S}_\alpha^\Omega = \left\{ S \in \mathbb{R}^{n_1 \times n_2} \middle| \forall i : \|\mathcal{P}_\Omega(S)_{(i,\cdot)}\|_0 \leq \alpha r_i \wedge \forall j : \|\mathcal{P}_\Omega(S)_{(\cdot,j)}\|_0 \leq \alpha c_j \right\}. \tag{5}$$

Assumption 1 was introduced by Candes and Recht (2012). Assumptions 2 and 3 or variants thereof are common in theoretical analyses of matrix completion (see for example, Candès et al., 2011; Yi et al., 2016; Cherapanamjeri et al., 2017; Wang and Wei, 2024).

Similar to other works on robust matrix completion, we derive guarantees for a slightly modified variant of our proposed method. Specifically, we make the following two modifications.

First, the set of corrupted entries $\Lambda_*$ is estimated differently. In the original method $\Lambda_*$ is estimated as the set of $k$ largest residual entries. However, motivated by Assumption 3 that the fraction of corrupted entries in each row and column is bounded, in the modified algorithm we remove only a bounded number of entries from each row and column. To this end we introduce the following thresholding operator $\mathcal{T}_\theta(A, \Omega)$. Given an input matrix $A$, a subset $\Omega$, and a parameter $\theta \in (0, 1)$, the operator $\mathcal{T}_\theta(A, \Omega)$ keeps only those entries $(i, j) \in \Omega$ which belong to the largest $\theta$-fraction of entries in *both* the respective row $i$ and column $j$, and zeros out all the remaining entries. In case of entries having identical magnitude, ties are broken arbitrarily. Formally,

$$\mathcal{T}_\theta(A, \Omega) = \begin{cases} \mathcal{P}_\Omega(A)_{i,j} & |A_{i,j}| > \left[|\mathcal{P}_\Omega(A)|_{(i,\cdot)}\right]^{(\lceil \theta r_i \rceil)} \wedge |A_{i,j}| > \left[|\mathcal{P}_\Omega(A)|_{(\cdot,j)}\right]^{(\lceil \theta c_j \rceil)} \\ 0, & \text{otherwise.} \end{cases} \tag{6}$$

At each iteration $t$, given the current estimate $L_t = U_t V_t^\top$, the estimated set of corrupted entries is

$$\Lambda_t = \text{support}\left(\mathcal{T}_\theta\left(L_t - X, \Omega\right)\right). \tag{7}$$

As in the original method, entries are still removed based on the magnitude of their residual but in this way the number of removed entries from each row and column is bounded. For RGNMR to be able to remove all outliers, a necessary condition is that $\theta \geq \alpha$. In what follows, we choose $\theta = \gamma\alpha$ where $\gamma \geq 1$ is the over-removal factor.

The second modification to RGNMR is to constrain the update $(U_{t+1}, V_{t+1})$ to be in the vicinity of the current pair of factor matrices $(U_t, V_t)$ and to have bounded row norms. Formally, for a parameter

$\delta_t > 0$ we define the $\delta_t$ neighborhood of the current estimate $(U_t, V_t)$ as

$$\mathcal{C}(U_t, V_t, \delta_t) = \left\{ \begin{pmatrix} \tilde{U} \\ \tilde{V} \end{pmatrix} \in \mathbb{R}^{(n_1+n_2)\times r} \,\middle|\, \|\tilde{U} - U_t\|_F^2 + \|\tilde{V} - V_t\|_F^2 \le \delta_t \right\}, \tag{8}$$

and denote the subset of factor matrices with bounded row norms by

$$\mathcal{B}_\mu = \left\{ \begin{pmatrix} U \\ V \end{pmatrix} \in \mathbb{R}^{(n_1+n_2)\times r} \,\middle|\, \|U\|_{2,\infty} \le \sqrt{\frac{3\mu r \sigma_1^*}{n_1}}, \quad \|V\|_{2,\infty} \le \sqrt{\frac{3\mu r \sigma_1^*}{n_2}} \right\}, \tag{9}$$

where the constant 3 is arbitrary. Instead of Eq. (1), for a suitably chosen $\delta > 0$, the modified RGNMR updates the factor matrices as follows,

$$(U_{t+1}, V_{t+1}) = \arg\min \left\{ \mathcal{L}^t_{\Omega\setminus\Lambda_t}(U, V) \,\middle|\, (U, V) \in \mathcal{B}_\mu \cap \mathcal{C}\left(U_t, V_t, \frac{\delta}{4^{t+1}}\right) \right\}. \tag{10}$$

Similar constraints were employed by Zilber and Nadler (2022) and Keshavan et al. (2010), in deriving theoretical guarantees for their (non-robust) matrix completion algorithms. As these constraints are quadratic, the above problem may be written as a convex optimization problem with quadratic regularization terms. Hence, it can be solved computationally efficiently. The modified RGNMR is described in Algorithm 2.

**Main theorems.** To state our recovery guarantees we introduce the following definitions. For a rank $r$ matrix $L^*$ with smallest singular value $\sigma_r^*$, we define the following sets, as in Zilber and Nadler (2022). First, we denote all factorizations of rank-$r$ matrices with a bounded error from $L^*$ by

$$\mathcal{B}_{\text{err}}(\epsilon) = \left\{ (U, V) \in \mathbb{R}^{(n_1+n_2)\times r} \,\middle|\, \|UV^\top - L^*\|_F \le \epsilon\sigma_r^* \right\}. \tag{11}$$

In particular, we denote by $\mathcal{B}^* = \mathcal{B}_{\text{err}}(0)$ the set of all exact factorizations of $L^*$,

$$\mathcal{B}^* = \left\{ (U, V) \in \mathbb{R}^{(n_1+n_2)\times r} \,\middle|\, UV^\top = L^* \right\}. \tag{12}$$

We say that $U, V$ are *balanced* if $U^\top U = V^\top V$, and measure the imbalance by $\|U^\top U - V^\top V\|_F$. We denote all the pairs of factor matrices which are approximately balanced by

$$\mathcal{B}_{\text{bln}}(\delta) = \left\{ (U, V) \in \mathbb{R}^{(n_1+n_2)\times r} \,\middle|\, \|U^\top U - V^\top V\|_F \le \delta\sigma_r^* \right\}. \tag{13}$$

Our first theorem states that starting from a sufficiently accurate balanced initialization with bounded row norms, the modified RGNMR of Algorithm 2 recovers $L^*$ with high probability and with a linear convergence rate.

**Theorem 3.1.** *Let $X = L^* + S^*$, where $L^* \in \mathcal{M}(n_1, n_2, r, \mu, \kappa)$, and without loss of generality $n_1 \ge n_2$. Let the set of observed entries $\Omega$ follow Assumption 2, and the corruption matrix $S^*$ satisfy Assumption 3 for some known $\alpha \in (0, 1)$. For large enough absolute constants $C, c_l, c_e, c_\alpha, c_\gamma$ the following holds: If the fraction of corrupted entries is small enough, $\alpha < \frac{1}{c_\alpha r\mu\kappa}$ and the probability to observe an entry is high enough, $p \ge \frac{C\mu r}{n_2} \max\{\log n_1, \mu r\kappa^2\}$, then w.p. at least $1 - \frac{6}{n_1}$, Algorithm 2 with parameter $\frac{25\sigma_r^*}{c_e^2\kappa} \le \delta \le \frac{\sigma_r^*}{c_e\kappa}$, over removal factor $c_\gamma \le \gamma \le \sqrt{c_\alpha}$ and an initialization $(U_0, V_0) \in \mathcal{B}_{\text{err}}(\frac{1}{c_e\sqrt{\kappa}}) \cap \mathcal{B}_{\text{bln}}(\frac{1}{2c_l}) \cap \mathcal{B}_\mu$ converges linearly to $L^*$. That is, after $t$ iterations the estimate $L_t = U_t V_t^\top$ satisfies*

$$\|L_t - L^*\|_F \le \frac{1}{2^t} \frac{\sigma_r^*}{c_e\sqrt{\kappa}}. \tag{14}$$

Our second theorem states that under suitable conditions it is possible to construct an accurate initialization that satisfies the requirements of Theorem 3.1. To do so we employ the spectral based initialization scheme proposed by Yi et al. (2016), followed by a normalization procedure on the rows of the factor matrices. Similar normalization procedures were employed by Zheng and Lafferty (2016) and Zilber and Nadler (2022) for their analysis of matrix completion without outliers. Our initialization scheme, outlined in Algorithm 3, is described in detail in Appendix A.

**Theorem 3.2.** *Let $X = L^* + S^*$, $L^* \in \mathcal{M}(n_1, n_2, r, \mu, \kappa)$, $S^* \in \mathcal{S}_\alpha^\Omega$ and without loss of generality, $n_1 \geq n_2$. Then for any $c_l, c_e$ for large enough constants $c_\alpha, C$ the following holds: If $\alpha \leq \frac{1}{c_\alpha \kappa^2 r^{\frac{3}{2}} \mu}$ and $p \geq C \frac{\mu r^2 \kappa^4 \log n_1}{n_2}$ then w.p. at least $1 - \frac{6}{n_1}$ Algorithm 3 outputs a pair $(U_0, V_0)$ that satisfies*

$$(U_0, V_0) \in \mathcal{B}_{\text{err}}\left(\frac{1}{c_e\sqrt{\kappa}}\right) \cap \mathcal{B}_{\text{bln}}\left(\frac{1}{2c_l}\right) \cap \mathcal{B}_\mu. \tag{15}$$

**Remark 3.1.** *Theorem 3.1 holds for $\frac{25\sigma_r^*}{c_e^2\kappa} \leq \delta \leq \frac{\sigma_r^*}{c_e\kappa}$. In principle, to provide a valid value of $\delta$ in this interval requires knowledge of $\sigma_1^*$ and $\sigma_r^*$. We note that several other works also required knowledge of $\sigma_1^*$ or $\sigma_r^*$ or both for their methods (see Yi et al., 2016; Sun and Luo, 2016; Cherapanamjeri et al., 2017). In fact, for our approach it suffices to estimate these two singular values up to constant multiplicative factors. Moreover, within the proof of Theorem 3.2, we show that under its assumptions, it is indeed possible to estimate $\sigma_1^*$ and $\sigma_r^*$ up to constant factors.*

**Remark 3.2.** *Theorems 3.1 and 3.2 require that the parameters $\mu$ and $\alpha$ are known. Similar assumptions were made in previous works (e.g. Yi et al., 2016; Cherapanamjeri et al., 2017; Sun and Luo, 2016; Zilber and Nadler, 2022).*

**Comparison to other recovery guarantees.** We compare our theoretical results to those derived for both factorization based methods as well as methods that operate on the full matrix.

For the vanilla matrix completion problem, where the observed entries are not corrupted, the smallest sample complexity requirement for factorization based methods was derived by Zheng and Lafferty (2016). Remarkably, Theorem 3.1 matches this result for `RGNMR` even in the presence of corrupted entries. The number of samples required by Theorem 3.1 is smaller than those required for other RMC methods (Yi et al., 2016; Wang and Wei, 2024) and if $\kappa < C \log(\mu r \sigma_1^*) \log n_1$, for a suitable constant $C$, it is also smaller than the requirements in Cherapanamjeri et al. (2017).

In terms of the fraction $\alpha$ of corrupted entries, the recovery guarantee with the highest $\alpha$ for factorization based methods was derived by Tong et al. (2021) and by Cai et al. (2024), under the assumption that the matrix is fully observed. Theorem 3.1 improves upon this result by a factor of $\mathcal{O}(\sqrt{r})$, even though the matrix is only partially observed. Hence our recovery guarantee holds under the weakest known conditions for factorization based methods. For general RMC methods the recovery guarantee with the highest $\alpha$ was derived by Cherapanamjeri et al. (2017). Theorem 3.1 requires a smaller $\alpha$ than in Cherapanamjeri et al. (2017) by a factor of $\mathcal{O}(\kappa)$. Table 1 summarizes the sample complexity and corruption rate requirements of different methods.

Next, for constructing a suitable initialization the requirements on $\alpha$ and $p$ in Theorem 3.2 are more stringent than those for the recovery guarantee of Theorem 3.1. This separation is common in other works on matrix completion (Zheng and Lafferty, 2016; Yi et al., 2016; Zilber and Nadler, 2022). In terms of sample complexity Theorem 3.2 still improves upon the results in Yi et al. (2016) and Wang and Wei (2024) and in some parameter regimes also upon the result in Cherapanamjeri et al. (2017). In terms of the fraction of outliers the condition in Theorem 3.2 is more stringent than in Yi et al. (2016) and Cherapanamjeri et al. (2017). Since empirically `RGNMR` succeeds even with random initialization we believe the condition imposed by Theorem 3.1 can be relaxed. This would allow for less stringent conditions in Theorem 3.2. We leave this for future work.

# 4 Numerical results

This section details the simulations conducted to compare the various RMC methods. The results are illustrated in Figures 1, 2, 3 and in Appendix B. Each instance consisted of generating a random matrix $L^* \in \mathbb{R}^{n_1 \times n_2}$ of a given rank $r$, condition number $\kappa$, subset of observed entries $\Omega \subseteq [n_1] \times [n_2]$ and a corruption matrix $S^*$. We generate $L^*$ following Tong et al. (2021). Specifically, we construct $U \in \mathbb{R}^{n_1 \times r}, V \in \mathbb{R}^{n_2 \times r}$ with entries i.i.d. from the standard normal distribution and orthonormalize their columns. We then construct $\Sigma \in \mathbb{R}^{r \times r}$ as a diagonal matrix with $r$ evenly spaced values between 1 and $\frac{1}{\kappa}$ on the diagonal. We set $L^* = U\Sigma V^\top$. Next, to generate $\Omega$ we follow Zilber and Nadler (2022). Given an oversampling ratio $\rho$ we sample $\Omega$ of size $\lfloor \rho r(n_1 + n_2 - r) \rfloor$ randomly without replacement and verify that it contains at least $r$ entries in each column and row. To construct the corruption matrix $S^*$ we first sample its support $\Lambda_* \subseteq \Omega$ as follows. Given a fraction $\alpha$ of corrupted entries we sample $\lfloor \alpha \cdot |\Omega| \rfloor$ entries out of $\Omega$ randomly without replacement. We then verify that the

set $\Omega \setminus \Lambda_*$ contains at least $r$ entries in each column and row. Similar to Cai et al. (2024) and Wang and Wei (2024), we sample the value of each corruption entry uniformly between $-\max_{i,j} |L_{i,j}^*|$ and $\max_{i,j} |L_{i,j}^*|$. Each method is then provided with the matrix $\mathcal{P}_\Omega(L^* + S^*)$, the set of entries $\Omega$ and the matrix rank $r$.

We compare `RGNMR` to `AOP` (Yan et al., 2013), `RMC` (Cambier and Absil, 2016), `RPCA-GD` (Yi et al., 2016), `RRMC` (Cherapanamjeri et al., 2017), `HUB` (Ruppel et al., 2020) and `HOAT` (Wang et al., 2023). In addition, as described in Section 2.1, we compare `RGNMR` to `RGNMR-BS`. Implementations of all methods except `RRMC` are publicly available. An implementation of `RRMC` was kindly provided to us by Wang and Wei (2024). MATLAB and Python implementations of `RGNMR` are available at `github.com/eilon96/RGNMR`. All simulations were run on a 2.1GHz Intel Core i7 CPU with 32GB of memory.

Each algorithm was executed with its default parameters, with the following exceptions: (i) To improve `RPCA-GD` and `HUB` performance we increased their number of iterations by using stricter convergence threshold than their default; (ii) In `RPCA-GD` we employed an over removal factor $\gamma$ of 4 as it significantly improved its performance in our simulations; (iii) We tuned the parameter $\xi$ in `HOAT` to 30; (iv) We provided `AOP` the exact number of corrupted entries; (v) For `RRMC` we use the same parameters used in Wang and Wei (2024).

In Appendix B we illustrate the performance of `RGNMR` under a broad set of additional scenarios. Specifically, Figure 6 shows that `RGNMR` can handle a large fraction of corrupted entries. Figures 7 and 8 show that `RGNMR` outperforms other methods also under non-uniform sampling. Figures 9, 10 and 11 illustrate its performance under a combination of additive noise and outliers. Figure 12 demonstrates that `RGNMR` can recover matrices with a high rank even if the number of observed entries is relatively small. Figure 13 illustrates that `RGNMR` performs well on a real dataset involving background extraction in a video. This is a standard benchmark for RMC methods (Yi et al., 2016; Cherapanamjeri et al., 2017; Huang et al., 2021; Cai et al., 2024). Finally, Figure 14 shows the runtime of `RGNMR` and illustrates its performance on large matrices.

## 5 Limitations and future research

In this section we discuss the theoretical and empirical limitations of `RGNMR`. These suggest several promising directions for future research.

On the theoretical front, though our analysis improves upon existing results, it does not fully account for the remarkable empirical performance of `RGNMR`. For example, the conditions in Theorem 3.1 on both $\alpha$ and $p$ depend on the condition number $\kappa$ while empirically the performance of `RGNMR` is not affected by $\kappa$. In addition, our analysis did not consider the case of overparameterization. These are open directions for future theoretical analysis of `RGNMR`.

On the empirical front, although `RGNMR` performs well in cases where most RMC methods fail, it is more computationally demanding than some of them. This is because each optimization step of `RGNMR` requires to solve a large system of linear equations (1), whereas some other methods require only a gradient descent step. As we illustrate in Figure 14, the runtime of `RGNMR` grows quadratically with $n$. Improving the runtime of `RGNMR` is therefore an interesting direction for future research.

Another possible line of research is to employ `RGNMR` to the problem of robust matrix recovery (Li et al., 2020b; Ding et al., 2021; Ma and Fattahi, 2023). As discussed in Remark 2.2, `RGNMR` is closely related to the `GNMR` method (Zilber and Nadler, 2022). Since `GNMR` was proven to successfully handle the problem of vanilla matrix recovery, where no data is corrupted, we believe the same can be achieved for `RGNMR` in the presence of corrupted data.

**Acknowledgments**    B.N. is the incumbent of the William Petschek Professorial Chair of Mathematics. The research of B.N. was supported in part by grant 2362/22 from the Israel Science Foundation. We thank Tianming Wang and Ke Wei for sharing the code of their work (Wang and Wei, 2024). We also thank the anonymous reviewers for their valuable feedback that improved the quality of our manuscript.

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

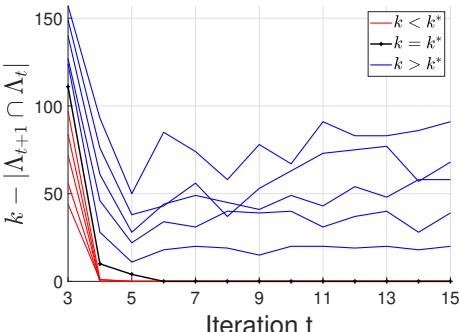

Figure 4: Empirical convergence or non-convergence of $\Lambda_t$ as a function of the iteration $t$ of `RGNMR` for $k < k^*$ (red), $k = k^*$ (black) or $k > k^*$ (blue). The assumed number of corrupted entries was in the range $k \in [0.95, 1.05] \cdot k^*$. As shown, if $k > k^*$ $\Lambda_t$ does not converge.

## A    Initialization procedure

Theorem 3.1 shows that starting from a sufficiently accurate balanced initialization with bounded row norms `RGNMR` recovers $L^*$. Therefore, the goal of our initialization procedure is to construct such initialization. Our proposed scheme starts by constructing an initial estimator of $S^*$, which we denote by $S_{init}$. To this end, the function $\mathcal{T}_\alpha$ (defined in eq. (6)) is applied to $\mathcal{P}_\Omega(X)$. Formally,

$$S_{init} = \mathcal{T}_\alpha \left( \mathcal{P}_\Omega(X), \Omega \right). \tag{16}$$

If the initial estimate of $S^*$ is accurate enough, $S_{init} \approx S^*$, then $\frac{1}{p}(\mathcal{P}_\Omega(X) - S_{init}) \approx \mathbb{E}[\mathcal{P}_\Omega(L^*)]$, should be a sufficiently accurate estimation of $L^*$. Denote by $\tilde{U}\tilde{\Sigma}\tilde{V}^\top$ the SVD of $P_r\left(\frac{1}{p}(\mathcal{P}_\Omega(X) - S_{init})\right)$. To obtain a balanced factorization of $\frac{1}{p}(\mathcal{P}_\Omega(X) - S_{init})$ we use the following operator,

$$\text{b-SVD}_r(\frac{1}{p}(\mathcal{P}_\Omega(X) - S_{init})) = \begin{pmatrix} \tilde{U}\tilde{\Sigma}^{\frac{1}{2}} \\ \tilde{V}\tilde{\Sigma}^{\frac{1}{2}} \end{pmatrix} = \begin{pmatrix} U \\ V \end{pmatrix}. \tag{17}$$

Note that the output of this operator is a perfectly balanced factorization which satisfies $U^\top U = V^\top V$. To bound the row norms of the factor matrices we apply a clipping operator $R_\eta$ with a suitably chosen $\eta$. The operator is applied to a vector $\mathbf{x}$ and it is defined as

$$R_\eta(\mathbf{x}) = \begin{cases} \mathbf{x} & \|\mathbf{x}\| \leq \eta, \\ \eta\frac{\mathbf{x}}{\|\mathbf{x}\|} & \|\mathbf{x}\| > \eta. \end{cases} \tag{18}$$

For a matrix $A$, we define $R_\eta(A)$ as the matrix obtained by applying $R_\eta$ to each of its rows. For every row $i$, $R_\eta(\mathbf{A})_{(i,\cdot)} = R_\eta(\mathbf{A}_{(i,\cdot)})$. Algorithm 3 outlines this initialization scheme.

## B    Additional simulations

We present here results of additional simulations under various settings beyond those described in the main text.

**Fraction of outliers.**    Figure 6 shows the performance of various RMC methods as a function of the fraction of corrupted entries $\alpha$. We only compare methods that performed well at an oversampling ratio of $\frac{|\Omega|}{r(n_1+n_2-r)} = 8$ with $\alpha = 5\%$, see Figure 1, excluding `RPCA-GD`, `HOAT` and `RRMC`. As shown in the left figure if the number of observed entries is relatively small, at oversampling ratio of $8$, then `RGNMR` can handle a larger fraction of corrupted entries than other methods. In the right figure we show that if the number of observed entries is relatively large, oversampling ratio of $12$, then methods such as `RMC` and `AOP` can handle a larger number of corrupted entries than `RGNMR`. We note that if the condition number is high then `RGNMR` still outperforms these methods.

---
**Algorithm 3** Initialization procedure for Robust Matrix Completion
---
**Input:**

- $\{X_{i,j} \mid (i,j) \in \Omega\}$ - observed entries
- $\mu$ incoherence parameter of $L^*$
- $r$ - rank of $L^*$
- $p$ - probability of observing an entry
- $\alpha$ - the maximal fraction of outliers in each row/column

**Output:** $(U_0, V_0)$

$\quad S_{init} = \mathcal{T}_\alpha(\mathcal{P}_\Omega(X), \Omega)$ // An estimate of $S^*$

$\quad$ set $Z = \begin{pmatrix} U \\ V \end{pmatrix} = \text{b-SVD}_r \left[ \frac{1}{p} \left( \mathcal{P}_\Omega(X) - S_{init} \right) \right]$

$\quad$ set $\eta_1 = \sqrt{\frac{\mu r}{n_1}} \|Z\|_{op}$ and $\eta_2 = \sqrt{\frac{\mu r}{n_2}} \|Z\|_{op}$

$\quad$ set $Z_0 = \begin{pmatrix} R_{\eta_1}(U_0) \\ R_{\eta_2}(V_0) \end{pmatrix}$

$\quad$ **return** $(U_0, V_0)$
---

**Non-uniform sampling.** In many applications, the entries of the matrix are not sampled uniformly (Meka et al., 2009; Okatani et al., 2011). Hence we made simulations where the observed entries followed a power law sampling scheme similar to Meka et al. (2009) and a diagonal-band pattern as in Okatani et al. (2011).

For the power law scheme, given an oversampling ratio $\rho$ we define $w = \rho \cdot r \cdot (n_1 + n_2 - r)$. We construct two sequences $(\tilde{p}_1 \ldots \tilde{p}_{n_1})$ and $(\tilde{q}_1 \ldots \tilde{q}_{n_2})$ such that $\tilde{p}_i = i^{-\frac{2}{3}}$, $\tilde{q}_i = i^{-\frac{2}{3}}$. We then normalize them to construct two new sequences

$$p_i = w \cdot \frac{\tilde{p}_i}{\sum_j \tilde{p}_j}, \quad q_i = w \cdot \frac{\tilde{q}_i}{\sum_j \tilde{q}_j}.$$

Note that $\sum_i p_i = \sum_j q_j = w$. We sample each entry $(i,j)$ with probability $\frac{p_i q_j}{w}$. The expected number of observed entries is then $\mathbb{E}[\Omega] = w$. In Figure 7 we illustrate that RGNMR performs better than most methods when the entries are sampled under this scheme.

For the diagonal-band pattern, we generated $n \times n$ matrices of rank $r$ with a diagonal bandwidth of length $pr$ across different values of $p$. This results in approximately $(n + n) \cdot pr$ observed entries and therefore $p$ is approximately the oversampling ratio. As illustrated in Figure 8, though RGNMR requires a larger oversampling ratio than in the uniform pattern to succeed in this task, it stills outperforms other RMC methods.

**Outliers and additive noise.** In Figures 9, 10 and 11 we illustrate that RGNMR still outperform most RMC methods even under additive noise. In these simulation, all observed entries are corrupted by additive white Gaussian noise with known standard deviation $\sigma$, in addition to the few outliers entries. In this scenario inliers entries are corrupted with random noise. Note that our method for upper bounding $k^*$ is based on the assumption that the error in those entries is the result of rounding errors, see 2.1. To overcome this problem, when searching for an upper bound on $k^*$ we terminated RGNMR once $\frac{\|L_t - X\|_{F(\Omega \setminus \Lambda_t)}}{\|X\|_{F(\Omega \setminus \Lambda_t)}} \leq \sqrt{\sigma}$.

**High rank matrices.** In Figure 12 we compare the performance of various RMC method as a function of the rank of the target matrix $L^*$. Following (Huang et al., 2021; Wang and Wei, 2024) we fix the corruption rate $\alpha = 0.1$ and vary the rank and the fraction of observed entries $\frac{|\Omega|}{n_1 n_2}$. As shown, RGNMR successfully recovers $L^*$ from a small fraction of the observed entries even if the rank of $L^*$ is relatively high.

**Real data: Background extraction from video.** In Figure 13 we illustrate RGNMR performance for video background extraction. We use the data from (Li et al., 2004), kindly provided to us by the

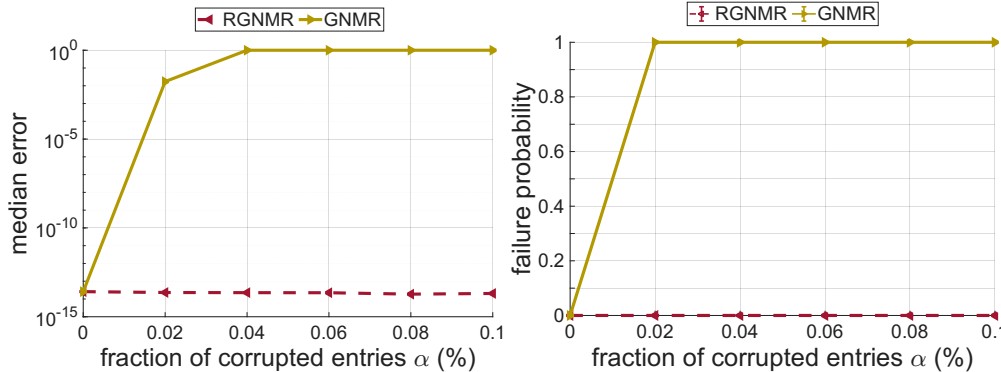

Figure 5: RGNMR performance compared to that of GNMR (Zilber and Nadler, 2022) under a small fraction of corrupted entries. (left) Median rel-RMSE ; (right) Failure probability defined as $\mathbb{P}(\text{rel-RMSE} > 10^{-3})$. The underlying matrix $L^*$ is of size $500 \times 500$ has rank $r = 5$ and condition number $\kappa = 2$. The oversampling ratio is $\frac{|\Omega|}{r(n_1+n_2-r)} = 12$ and the fraction of corrupted entries $\alpha$ varies. Each point corresponds to 50 independent realizations.

authors of (Huang et al., 2021). The data contains a sequence of grayscale frames. By stacking the columns of each frame of the video into a long vector, we obtain a matrix whose columns correspond to the frames. This matrix can be decomposed to a low rank matrix corresponding to the static background plus a sparse matrix corresponding to the moving foreground. Following (Yi et al., 2016; Cherapanamjeri et al., 2017; Huang et al., 2021; Cai et al., 2024) we sample uniformly at random $5\%$ of the matrix entries. All method are then given the sampled entries and an input rank of $r = 1$. As shown RGNMR performs well on real data and successfully extracts the background.

**Runtime.** Finally, In Figure 14 we illustrate that RGNMR scales well with the matrix size. We show that for a matrix of size $n \times n$ the runtime of RGNMR and RGNMR-BS increases quadratically with $n$. In addition, we illustrate that even on large matrices RGNMR still requires a relatively small number of observed entries to succeed.

## C  Proofs of Theorems 3.1 and 3.2

To prove Theorems 3.1 and 3.2, we make use of several auxiliary lemmas. These are outlined in subsections C.1 and C.3. The proofs of the two theorems appear in subsections C.2 and C.4.

### C.1  Auxiliary Lemmas For Theorem 3.1

The first three lemmas are results from (Zilber and Nadler, 2022). The first one is a combination of Lemmas SM5.3 and SM5.5 from (Zilber and Nadler, 2022).

**Lemma C.1.** *Let $L^* \in \mathcal{M}(n_1, n_2, r, \mu, \kappa)$ and let $\Omega$ follow Assumption 2 with probability $p$. Let $\epsilon \in (0,1)$. There exist constants $C, c_l, c_e$ such that the following holds: If $p \geq C \max\{\frac{\log n_1}{n_2}, \frac{\mu^2 r^2 \kappa^2}{n_2 \epsilon^4}\}$. Then w.p. at least $1 - \frac{2}{n_1^5}$, for any $(U, V) \in \mathcal{B}_{\text{err}}(\epsilon/c_e) \cap \mathcal{B}_{\text{bln}}(1/c_l) \cap \mathcal{B}_\mu$ with $L = UV^\top$ there exists $(U^*, V^*) \in \mathcal{B}^* \cap \mathcal{B}_\mu$ such that*

$$\|U - U^*\|_F^2 + \|V - V^*\|_F^2 \leq \frac{25}{4\sigma_r^*}\|UV^\top - L^*\|_F^2, \tag{19a}$$

$$\frac{1}{\sqrt{p}}\|(U - U^*)(V - V^*)^\top\|_{F(\Omega)} \leq \frac{\epsilon}{6}\|L - L^*\|_F. \tag{19b}$$

The second lemma is Lemma SM5.4 in (Zilber and Nadler, 2022).

**Lemma C.2** (uniform RIP for matrix completion). *Let $L^* \in \mathcal{M}(n_1, n_2, r, \mu, \kappa)$ and let $\Omega$ follow Assumption 2 with probability $p$. Let $\epsilon \in (0,1)$. There exist constants $C, c_l, c_e$ such that the*

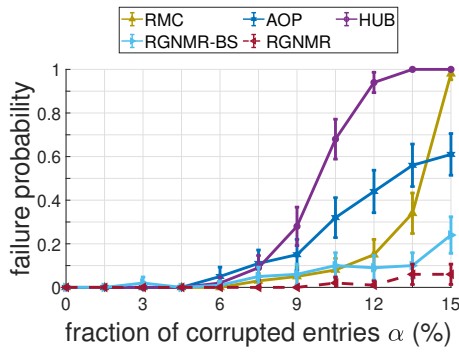 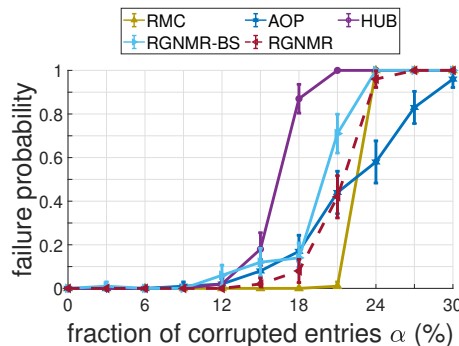

Figure 6: Failure probability ($\pm1.96$SE) of RMC methods as a function of the fraction of corrupted entries $\alpha$. (left) Oversampling ratio of $\frac{|\Omega|}{r(n_1+n_2-r)} = 8$; (right) Oversampling ratio of $\frac{|\Omega|}{r(n_1+n_2-r)} = 12$. The underlying matrix $L^*$ if of size $3200 \times 400$ has a rank $r = 5$ and a condition number $\kappa = 2$. Each point corresponds to 100 independent realizations.

following holds: If $p \geq C\frac{\log n_1 \mu r}{n_2 \epsilon^2}$. Then w.p. at least $1 - \frac{3}{n_1^3}$, for all matrices $L = UV^\top$ where $(U, V) \in \mathcal{B}_{\mathrm{err}}(\epsilon/c_e) \cap \mathcal{B}_{\mathrm{bln}}(1/c_l) \cap \mathcal{B}_\mu$, the following holds,

$$(1-\epsilon)\|L - L^*\|_F \leq \tfrac{1}{\sqrt{p}}\|\mathcal{P}_\Omega(L - L^*)\| \leq (1+\epsilon)\|L - L^*\|_F. \tag{20}$$

The third lemma is Lemma SM5.6 in (Zilber and Nadler, 2022), which in turn is a direct consequence of Lemma 7.1 in (Keshavan et al., 2010). In (Zilber and Nadler, 2022) the result is stated with an unspecified constant $c$. Tracking its proof it can be shown that it holds with $c = 2$, which is how the lemma is stated below.

**Lemma C.3.** *Let $\Omega$ follow Assumption 2 with probability $p$. There exist a constant $C$ such that the following holds for any $\mu, t, \epsilon > 0$: If $p \geq C \max\{\frac{\log n_1}{n_2}, \frac{\mu^2 r^2}{n_2 \epsilon^4}\}$. Then w.p. at least $1 - \frac{2}{n_1^5}$, for any $(U, V) \in \mathbb{R}^{(n_1+n_2)\times r}$ such that*

$$\|U\|_{2,\infty} \leq 4\sqrt{\mu r t/n_1}, \quad \|V\|_{2,\infty} \leq 4\sqrt{\mu r t/n_2}, \tag{21}$$

*we have*

$$\frac{1}{p}\|UV^\top\|_{F(\Omega)}^2 \leq \frac{\|U\|_F^2 + \|V\|_F^2}{2}\left(2\left(\|U\|_F^2 + \|V\|_F^2\right) + t\epsilon^2\right). \tag{22}$$

The following two lemmas provide bounds on the estimates constructed by RGNMR at each iteration. The first lemma states that starting from an approximately balanced and sufficiently accurate estimate of $L^*$, if each new pair of factor matrices is sufficiently close to the previous one, then all factor matrices continue to be approximately balanced, and remain not too far from $L^*$. Its proof appears in Appendix D.1.

**Lemma C.4.** *Let $L^* \in \mathcal{M}(n_1, n_2, r, \mu, \kappa)$ with $r$-th singular value $\sigma_r^*$. For large enough constants $c_l, c_e$ the following holds: If $(U_0, V_0) \in \mathcal{B}_{\mathrm{err}}(\frac{1}{c_e\sqrt{\kappa}}) \cap \mathcal{B}_{\mathrm{bln}}(\frac{1}{2c_l}) \cap \mathcal{B}_\mu$ and for every $1 \leq s \leq t+1$, $(U_s, V_s) \in \mathcal{C}(U_{s-1}, V_{s-1}, \frac{\sigma_r^*}{4^s c_e \kappa})$, then $(U_{t+1}, V_{t+1}) \in \mathcal{B}_{\mathrm{err}}(\frac{13}{\sqrt{c_e}}) \cap \mathcal{B}_{\mathrm{bln}}(\frac{1}{c_l})$.*

The second lemma bounds the distance between $L^*$ and the updated estimate $L_{t+1}$, as a function of various quantities of the current estimate $L_t$. Its proof appears in Appendix D.2.

**Lemma C.5.** *Let $L_t = U_t V_t^\top$ be the estimate of $L^*$ at iteration $t$. Let $L_{t+1} = U_{t+1}V_{t+1}^\top$ be the updated estimate, where $(U_{t+1}, V_{t+1})$ are computed by (10) with some $\delta > 0$. Assume that $L_t$ is sufficiently close to $L^*$, so that the set $\mathcal{A}_t = \mathcal{B}^* \cap \mathcal{B}_\mu \cap \mathcal{C}\left(U_t, V_t, \frac{\delta}{4^{t+1}}\right)$ is non empty. Let $(U^*, V^*) \in \mathcal{A}_t$ and denote $\Delta U_t^* = U_t - U^*$, $\Delta U_{t+1} = U_{t+1} - U_t$ with similar definitions for $\Delta V_t^*$ and $\Delta V_{t+1}$. Then, the error on the non removed entries $\Omega_t = \Omega \setminus \Lambda_t$ of the updated estimate satisfies*

$$\begin{aligned}
\|L_{t+1} - L^*\|_{F(\Omega_t)} \leq &\sqrt{2}\|\Delta U_t^* \Delta V_t^{*\top}\|_{F(\Omega_t)} + \sqrt{2}\|\Delta U_{t+1}\Delta V_{t+1}^\top\|_{F(\Omega_t)} \\
&+ (1+\sqrt{2})\|S^*\|_{F(\Omega_t \cap \Lambda_*)}.
\end{aligned} \tag{23}$$

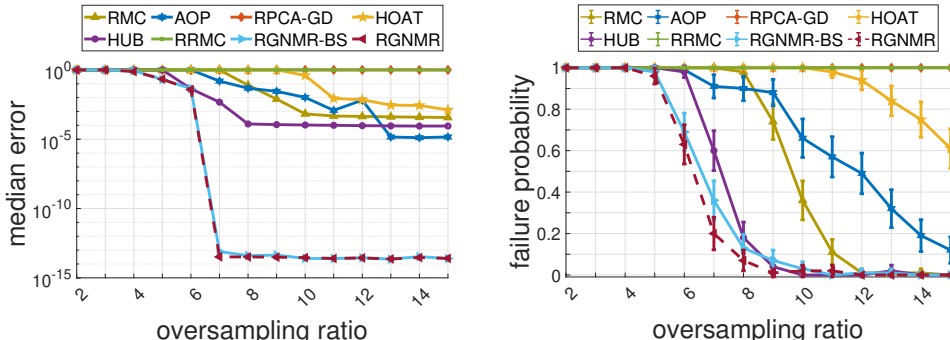

Figure 7: Results of RMC methods as a function of the oversampling ratio under power law sampling. (left) Median `rel-RMSE` ; (right) Failure probability defined as $\mathbb{P}(\texttt{rel-RMSE} > 10^{-3})$. The underlying matrix $L^*$ is of size $1000 \times 1000$ has rank $r = 5$ and condition number $\kappa = 2$. The fraction of corrupted entries is $\alpha = 5\%$. Each point corresponds to 50 independent realizations.

Before we present the last two lemmas, whose proofs appears in Appendix D.3, recall that at each iteration $t$, RGNMR removes from $\Omega$ the entries in $\Lambda_t = \operatorname{support}(\mathcal{T}_{\gamma\alpha}(L_t - X, \Omega))$. If RGNMR was unable to identify all the corrupted entries, then the set $(\Omega \setminus \Lambda_t) \cap \Lambda_*$, where $\Lambda_* = \operatorname{support}(S^*)$, is not empty. The following lemma bounds the magnitude of the entries of $S^*$ in $(\Omega \setminus \Lambda_t) \cap \Lambda_*$, for matrices $L_t$ that satisfy a suitable condition.

**Lemma C.6.** *Let $X = L^* + S^*$ where $L^* \in \mathcal{M}(n_1, n_2, r, \mu, \kappa)$ and the corruption matrix $S^*$ satisfies Assumption 3 for some known $0 < \alpha < 1$. Let the set of observed entries $\Omega$ follow Assumption 2 with $p \geq C\frac{\log n_1}{n_2}$ for some constant $C$. Let $L = UV^\top$ be an estimate of $L^*$ such that $(U, V) \in \mathcal{B}_\mu$. Suppose that the set of corrupted entries is estimated by $\Lambda = \operatorname{support}(\mathcal{T}_{\gamma\alpha}(L - X, \Omega))$ with an over removal factor of $1 < \gamma \leq \frac{1}{\alpha}$. If there exists a factorization $(U^*, V^*) \in \mathcal{B}^* \cap \mathcal{B}_\mu$ of $L^*$ such that*

$$\|U - U^*\|_F^2 + \|V - V^*\|_F^2 \leq \frac{25}{4\sigma_r^*}\|L - L^*\|_F^2, \tag{24}$$

*then the magnitude of the remaining corrupted entries in $(\Omega \setminus \Lambda) \cap \Lambda^*$, is bounded as follows,*

$$\|S^*\|_{F((\Omega\setminus\Lambda)\cap\Lambda_*)} \leq 18\sqrt{p\alpha\mu r\kappa}\|L - L^*\|_F + \sqrt{\frac{2}{\gamma - 1}}\|L - L^*\|_{F(\Omega)}. \tag{25}$$

The last lemma bounds the error term $\|L - L^*\|_{F(\Lambda_t)}$ for a matrix $L$ that satisfies a suitable condition.

**Lemma C.7.** *Let $L^* \in \mathcal{M}(n_1, n_2, r, \mu, \kappa)$ and let the set of observed entries $\Omega$ follow Assumption 2 with $p \geq C\frac{\log n_1}{n_2}$ for some constant $C$. Let $L = UV^\top$ be an estimate of $L^*$ such that $(U, V) \in \mathcal{B}_\mu$ and assume there exists a factorization $(U^*, V^*) \in \mathcal{B}^* \cap \mathcal{B}_\mu$ of $L^*$. If $\Lambda = \operatorname{support}(\mathcal{T}_{\gamma\alpha}(A, \Omega))$, for some matrix $A$, then*

$$\|L - L^*\|_{F(\Lambda)} \leq 27\sqrt{3}\gamma\alpha p\mu r\sigma_1^*(\|U - U^*\|_F^2 + \|V - V^*\|_F^2). \tag{26}$$

## C.2 Proof of Theorem 3.1

The proof relies on lemmas C.1-C.7 above. We first note that since $\Omega$ follows Assumption 2 with $p \geq \frac{C\mu r}{n_2}\max\{\log n_1, \mu r\kappa^2\}$, we may apply these lemmas. Specifically, for a large enough $C$ the conditions required by C.1 and C.2 with $\epsilon \geq \frac{1}{20}$ hold and the condition of Lemma C.3 holds with $\epsilon \geq \frac{1}{8\sqrt{c_e\kappa}}$.

*Proof of Theorem 3.1.* We prove by induction on the iteration step $t$ that

$$(U_t, V_t) \in \mathcal{B}_{\mathrm{err}}\left(\frac{1}{2^t c_e \sqrt{\kappa}}\right) \cap \mathcal{B}_{\mathrm{bln}}\left(\frac{1}{c_l}\right) \cap \mathcal{B}_\mu. \tag{27}$$

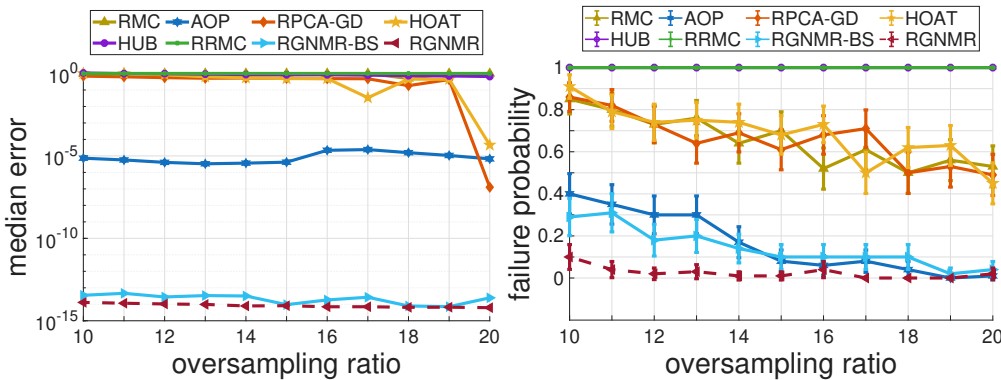

Figure 8: Results of RMC methods as a function of the oversampling ratio under diagonal-band pattern sampling. (left) Median `rel-RMSE`; (right) Failure probability defined as $\mathbb{P}(\texttt{rel-RMSE} > 10^{-3})$. The underlying matrix $L^*$ is of size $500 \times 500$ has rank $r = 5$ and condition number $\kappa = 2$. The fraction of corrupted entries is $\alpha = 5\%$. Each point corresponds to 50 independent realizations.

Eq. eq. (14) then follows immediately by the definition of $\mathcal{B}_{\text{err}}$, see eq. (11).

Eq.(27) for $t = 0$ follows from the assumption that the initialization is sufficiently accurate, balanced and has bounded row norms $(U_0, V_0) \in \mathcal{B}_{\text{err}}(\frac{1}{c_e\sqrt{\kappa}}) \cap \mathcal{B}_{\text{bln}}(\frac{1}{2c_l}) \cap \mathcal{B}_\mu$. Next we assume $(U_t, V_t)$ satisfies eq. (27) and prove that the updated matrices also satisfy this equation. Since at all intermediate steps $1 \leq s \leq t+1$, $(U_s, V_s)$ are updated by eq. (10), $(U_s, V_s) \in \mathcal{C}(U_{s-1}, V_{s-1}, \frac{\sigma_r^*}{4^s c_e \kappa}) \cap \mathcal{B}_\mu$. Hence by Lemma C.4, $(U_{t+1}, V_{t+1}) \in \mathcal{B}_{\text{bln}}(\frac{1}{c_l}) \cap \mathcal{B}_\mu$.

It remains to prove that $(U_{t+1}, V_{t+1}) \in \mathcal{B}_{\text{err}}(\frac{1}{2^{t+1} c_e \sqrt{\kappa}})$, namely that $\|L_{t+1} - L^*\|_F \leq \frac{\sigma_r^*}{2^{t+1} c_e \sqrt{\kappa}}$. The proof consist of two parts. First we bound the error restricted to the set $\Omega_t = \Omega \setminus \Lambda_t$, i.e. $\|L_{t+1} - L^*\|_{F(\Omega_t)}$. In the second part we use this result to bound the error on the entire matrix.

For the first part, to bound $\|L_{t+1} - L^*\|_{F(\Omega_t)}$ we apply Lemma C.5. To this end we first prove its conditions hold. In particular that the set $\mathcal{A}_t = \mathcal{B}^* \cap \mathcal{B}_\mu \cap \mathcal{C}(U_t, V_t, \frac{\delta}{4^{t+1}})$ is non empty. Indeed this follows by applying Lemma C.1. Specifically, since $(U_t, V_t) \in \mathcal{B}_{\text{err}}(\frac{1}{2^t c_e \sqrt{\kappa}}) \cap \mathcal{B}_{\text{bln}}(\frac{1}{c_l}) \cap \mathcal{B}_\mu$, for a large enough $c_e$, $(U_t, V_t)$ satisfy the conditions of Lemma C.1 with $\epsilon = \frac{1}{20}$. Consequently, w.p. at least $1 - \frac{2}{n_1^5}$, there exists a pair $(U^*, V^*) \in \mathcal{B}^* \cap \mathcal{B}_\mu$ that satisfy eqs. (19a) and (19b) with $U = U_t, V = V_t$. Combining eq. (19a) with the assumptions $\frac{25\sigma_r^*}{c_e^2 \kappa} \leq \delta$ and $\|L_t - L^*\| \leq \frac{\sigma_r^*}{2^t c_e \sqrt{\kappa}}$ yields

$$\|U_t - U^*\|_F^2 + \|V_t - V^*\|_F^2 \leq \frac{25}{4\sigma_r^*}\|L_t - L^*\|_F^2 \leq \frac{25}{4\sigma_r^*}\frac{\sigma_r^{*2}}{4^t c_e^2 \kappa} \leq \frac{\delta}{4^{t+1}}.$$

Therefore $(U^*, V^*) \in \mathcal{B}^* \cap \mathcal{B}_\mu \cap \mathcal{C}(U_t, V_t, \frac{\delta}{4^{t+1}})$.

We can now apply Lemma C.5 which gives

$$\|L_{t+1} - L^*\|_{F(\Omega_t)} \leq \sqrt{2}\left(\underbrace{\|\Delta U_t^* \Delta V_t^*\|_{F(\Omega_t)}}_{T_1} + \underbrace{\|\Delta U_{t+1} \Delta V_{t+1}^\top\|_{F(\Omega_t)}}_{T_2}\right) \tag{28}$$
$$+ (1 + \sqrt{2})\underbrace{\|S^*\|_{F(\Omega_t \cap \Lambda_*)}}_{T_3}.$$

We upper bound the term $T_1, T_2$ and $T_3$. The first term $T_1$ is bounded by eq. (19b) of Lemma C.1 with $\epsilon = \frac{1}{20}$.

$$T_1 = \|\Delta U_t^* \Delta V_t^*\|_{F(\Omega_t)} \leq \|\Delta U_t^* \Delta V_t^*\|_{F(\Omega)} \leq \frac{\sqrt{p}}{120}\|L_t - L\|_F \leq \frac{\sqrt{p}\sigma_r^*}{2^{t+6} c_e \sqrt{\kappa}}. \tag{29}$$

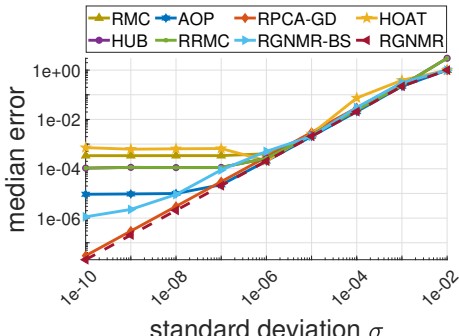

Figure 9: Performance of RMC methods in the presence of additive noise. The y-axis is the median `rel-RMSE`. The x-axis is the noise standard deviation. The underlying matrix $L^*$ is of size $3200 \times 400$ has rank $r = 5$ and condition number $\kappa = 2$. The fraction of corrupted entries is $\alpha = 5\%$. Each point corresponds to 50 independent realizations.

To bound $T_2$ we note that

$$\left(\frac{1}{\sqrt{p}} T_2\right)^2 = \frac{1}{p} \|\Delta U_{t+1} \Delta V_{t+1}^\top\|_{F(\Omega_t)}^2 \leq \frac{1}{p} \|\Delta U_{t+1} \Delta V_{t+1}^\top\|_{F(\Omega)}^2$$

To bound the above term we apply Lemma C.3. We first show that $(\Delta U_{t+1}, \Delta V_{t+1})$ satisfies eq. (21). Since $(U_{t+1}, V_{t+1})$ and $(U_t, V_t)$ are in $\mathcal{B}_\mu$ then

$$\|\Delta U_{t+1}\|_{2,\infty} \leq \|U_{t+1}\|_{2,\infty} + \|U_t\|_{2,\infty} \leq 2\sqrt{\frac{3\mu r \sigma_r^*}{n_1}} \leq 4\sqrt{\frac{\mu r \sigma_r^*}{n_1}},$$

and a similar inequality holds for $\Delta V_{t+1}$. Hence eq. (21) holds with $t = \sigma_r^*$. Consequently by eq. (22) of C.3 with $\epsilon = \frac{1}{8\sqrt{c_e \kappa}}$, w.p. at least $1 - \frac{2}{n_1^5}$

$$\frac{1}{p} \|\Delta U_{t+1} \Delta V_{t+1}^\top\|_{F(\Omega)}^2 \leq \frac{\|\Delta U_{t+1}\|_F^2 + \|\Delta V_{t+1}\|_F^2}{2} \left[2\left(\|\Delta U_{t+1}\|_F^2 + \|\Delta V_{t+1}\|_F^2\right) + \frac{\sigma_r^*}{64 c_e \kappa}\right].$$

Next we explicitly bound $\|\Delta U_{t+1}\|_F^2 + \|\Delta V_{t+1}\|_F^2$. Since $(U_{t+1}, V_{t+1}) \in \mathcal{C}\left(U_t, V_t, \frac{\delta}{4^{t+1}}\right)$, $\|\Delta U_{t+1}\|_F^2 + \|\Delta V_{t+1}\|_F^2 \leq \frac{\delta}{4^{t+1}}$. Hence

$$T_2 \leq \sqrt{p}\sqrt{\frac{\delta}{2 \cdot 4^{t+1}}\left[2\frac{\delta}{4^{t+1}} + \frac{\sigma_r^*}{64 c_e \kappa}\right]} \leq \sqrt{p}\sqrt{\frac{\delta^2}{2 \cdot 4^{2t+2}} + \frac{\delta \sigma_r^*}{128 c_e \cdot 4^{t+1} \kappa}}$$

Since $\delta \leq \frac{\sigma_r^*}{c_e \kappa}$

$$T_2 \leq \sqrt{p}\sqrt{\frac{\sigma_r^{*2}}{c_e^2 \kappa^2 \cdot 2^{4t+4}} + \frac{\sigma_r^{*2}}{c_e^2 \kappa^2 \cdot 2^{2t+9}}} \leq \sqrt{p}\sqrt{\frac{\sigma_r^{*2}}{c_e^2 \kappa^2 \cdot 2^{2t+3.8}}} \leq \frac{\sqrt{p}\sigma_r^*}{c_e \sqrt{\kappa} \cdot 2^{t+1.9}}. \tag{30}$$

In order to bound $T_3$ we invoke Lemma C.6. The condition of this Lemma, eq. (24), holds since we proved that $(U_t, V_t)$ satisfy eq. (19a) of Lemma C.1. Since $\Lambda_t = \text{support}\left(\mathcal{T}_\theta\left(L_t - X, \Omega\right)\right)$ we can employ Lemma C.6 which gives

$$T_3 \leq 18\sqrt{p\alpha\mu r\kappa}\|L_t - L^*\|_F + \sqrt{\frac{2}{\gamma - 1}}\|L_t - L^*\|_{F(\Omega)}. \tag{31}$$

Since $(U_t, V_t) \in \mathcal{B}_{\text{err}}\left(\frac{1}{2^t c_e \sqrt{\kappa}}\right)$, for a large enough $c_e$, $(U_t, V_t)$ satisfy the conditions of Lemma C.2 with $\epsilon = \frac{1}{20}$. Therefore by eq. (20) w.p. at least $1 - \frac{3}{n_1^3}$,

$$\sqrt{\frac{2}{\gamma - 1}}\|L_t - L^*\|_{F(\Omega)} \leq (1 + \frac{1}{20})\sqrt{\frac{2p}{\gamma - 1}}\|L_t - L^*\|_F \leq \sqrt{\frac{3p}{\gamma - 1}}\|L_t - L^*\|_F. \tag{32}$$

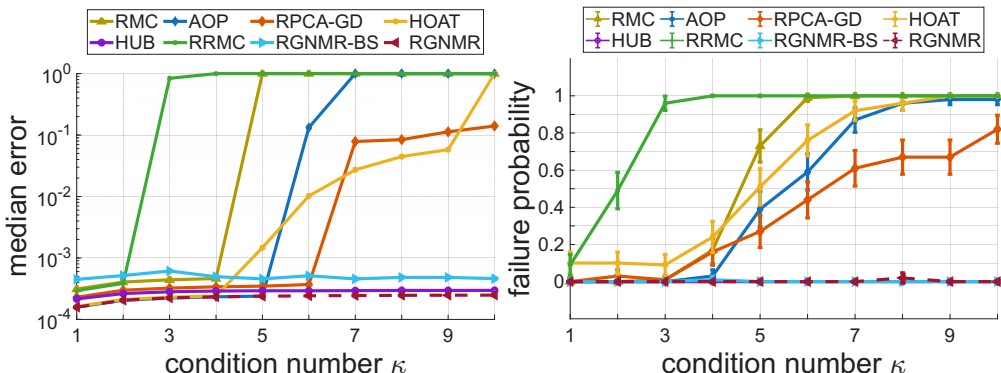

Figure 10: Performance of RMC methods as a function of the condition number under both outliers and additive noise. (left) Median `rel-RMSE`; (right) Failure probability ($\pm 1.96$ SE). The underlying matrix $L^*$ is of size $3200 \times 400$ has rank $r = 5$, the fraction of corrupted entries is $\alpha = 5\%$, the standard deviation of the white Gaussian noise is $\sigma = 10^{-6}$ and the oversampling ratio is $\frac{|\Omega|}{r \cdot (n_1 + n_2 - r)} = 12$. Each point corresponds to 100 independent realizations.

Combining eq. (31) and eq. (32) gives,

$$T_3 \leq \sqrt{p} \left( 18\sqrt{\alpha\mu r\kappa} + \sqrt{\frac{3}{\gamma - 1}} \right) \|L_t - L^*\|_F.$$

Since $\alpha < \frac{1}{c_\alpha \mu r\kappa}, c_\gamma \leq \gamma$ and by the induction hypothesis $\|L_t - L^*\|_F \leq \frac{\sigma_r^*}{2^t c_e \sqrt{\kappa}}$, then

$$T_3 \leq \sqrt{p} \left( 18\sqrt{\alpha\mu r\kappa} + \sqrt{\frac{3}{\gamma - 1}} \right) \frac{\sigma_r^*}{2^t c_e \sqrt{\kappa}} \leq \sqrt{p} \left( \frac{18}{\sqrt{c_\alpha}} + \sqrt{\frac{3}{c_\gamma - 1}} \right) \frac{\sigma_r^*}{2^t c_e \sqrt{\kappa}}.$$

For large enough $c_\alpha$ and $c_\gamma$, this yields

$$T_3 \leq \frac{\sqrt{p}\sigma_r^*}{2^{t+5} c_e \sqrt{\kappa}(1 + \sqrt{2})}. \tag{33}$$

Inserting the bounds on $T_1, T_2$ and $T_3$ from equations (29), (30) and (33) into eq. (28) gives

$$\|L_{t+1} - L^*\|_{F(\Omega_t)} \leq \frac{\sqrt{p}\sigma_r^*}{2^{t+1} c_e \sqrt{\kappa}} \left( \frac{1}{2^{4.5}} + \frac{1}{2^{0.4}} + \frac{1}{2^4} \right) \leq \frac{9}{10} \frac{\sqrt{p}\sigma_r^*}{2^{t+1} c_e \sqrt{\kappa}}. \tag{34}$$

Eq. (34) provides a bound on $L_{t+1} - L^*$, but only on the set $\Omega_t$. In what follows we show that this is sufficient for bounding the overall error on all entries, $\|L_{t+1} - L\|_F$. Since $\Omega_t = \Omega \setminus \Lambda_t$, then

$$\|L_{t+1} - L^*\|_{F(\Omega_t)}^2 = \|L_{t+1} - L^*\|_{F(\Omega)}^2 - \|L_{t+1} - L^*\|_{F(\Lambda_t)}^2. \tag{35}$$

To upper bound $\|L_{t+1} - L^*\|_{F(\Lambda_t)}^2$ we apply Lemma C.7, with $(U_{t+1}, V_{t+1})$ which indeed by their definition in eq. (10), belong to $B_\mu$, and with a specific factorization $(U^*, V^*)$ specified below. Specifically, we show that there exists a pair $(U^*, V^*) \in \mathcal{B}^* \cap \mathcal{B}_\mu$ that is in the vicinity of $(U_{t+1}, V_{t+1})$. By Lemma C.4 $(U_{t+1}, V_{t+1}) \in \mathcal{B}_{\text{err}}(\frac{13}{\sqrt{c_e}}) \cap \mathcal{B}_{\text{bln}}(\frac{1}{c_l}) \cap \mathcal{B}_\mu$. Therefore, for a large enough $c_e$ Lemma C.1 guarantees the existence of a factorization $(U^*, V^*) \in \mathcal{B}^* \cap \mathcal{B}_\mu$ that also satisfy eq. (19a) with $L = L_{t+1} = U_{t+1}V_{t+1}^\top$. Hence, by eq. (26) of Lemma C.7

$$\|L_{t+1} - L^*\|_{F(\Lambda_t)}^2 \leq 27\sqrt{3}\gamma\alpha p\mu r\sigma_1^*(\|U_{t+1} - U^*\|_F^2 + \|V_{t+1} - V^*\|_F^2). \tag{36}$$

Combining eq. (19a) and the assumptions $\alpha \leq \frac{1}{c_\alpha r\mu\kappa}, \gamma \leq \sqrt{c_\alpha}$ gives

$$\|L_{t+1} - L^*\|_{F(\Lambda_t)}^2 \leq \left( 27\sqrt{3}\gamma p\mu r \right) \frac{1}{c_\alpha \mu r\kappa} \frac{25}{4} \frac{\sigma_1^*}{\sigma_r^*} \|L_{t+1} - L^*\|_F^2$$

$$= \left( 27\sqrt{3}p \right) \frac{25}{4\sqrt{c_\alpha}} \|L_{t+1} - L^*\|_F^2. \tag{37}$$

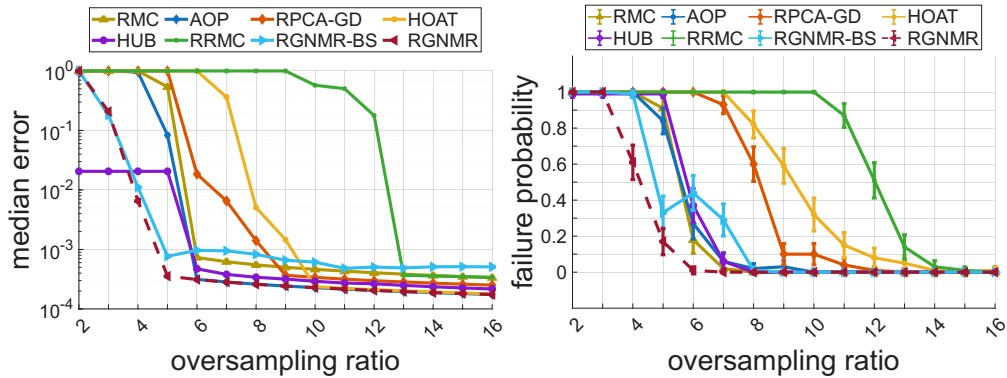

Figure 11: Performance of RMC methods as a function of the oversampling ratio under both outliers and additive noise. (left) Median `rel-RMSE`; (right) Failure probability ($\pm 1.96$ SE). The underlying matrix $L^*$ is of size $3200 \times 400$ has rank $r = 5$, the fraction of corrupted entries is $\alpha = 5\%$ the standard deviation of the white Gaussian noise is $\sigma = 10^{-6}$ and the condition number is $\kappa = 2$. Each point corresponds to 100 independent realizations.

Next we lower bound $\|L_{t+1} - L^*\|^2_{F(\Omega)}$ we apply Lemma C.2. Recall that by Lemma C.4, $(U_{t+1}, V_{t+1}) \in \mathcal{B}_{\mathrm{err}}(\frac{13}{\sqrt{c_e}}) \cap \mathcal{B}_{\mathrm{bln}}(\frac{1}{c_l}) \cap \mathcal{B}_\mu$. Hence, for a large enough $c_e$, $(U_{t+1}, V_{t+1})$ satisfies the condition of Lemma C.2, with $\epsilon = \frac{1}{20}$. By Lemma C.2, w.p. at least $1 - \frac{3}{n_1^3}$

$$\|L_{t+1} - L^*\|^2_{F(\Omega)} \geq (1 - \tfrac{1}{20})^2 p \|L_{t+1} - L^*\|^2_F \tag{38}$$

Inserting eq. (37) and eq. (38) into eq. (35) yields

$$\|L_{t+1} - L^*\|^2_{F(\Omega_t)} \geq p \left( \left(\tfrac{95}{100}\right)^2 - \tfrac{27\sqrt{3} \cdot 25}{4\sqrt{c_\alpha}} \right) \|L_{t+1} - L^*\|^2_F \geq p \left(\tfrac{9}{10}\right)^2 \|L_{t+1} - L^*\|^2_F$$

where the last inequality follows for a large enough $c_\alpha$. We conclude that

$$\|L_{t+1} - L^*\|_F \leq \frac{10}{9\sqrt{p}} \sqrt{p} \|L_{t+1} - L^*\|_{F(\Omega \backslash \Lambda_t)}.$$

By eq. (34) the right hand size is upper bounded by $\frac{\sigma_r^*}{2^{t+1} c_e \sqrt{\kappa}}$, which implies that $(U_{t+1}, V_{t+1}) \in \mathcal{B}_{\mathrm{err}}\left(\frac{1}{2^{t+1} c_e \sqrt{\kappa}}\right)$. Since $(U_{t+1}, V_{t+1}) \in \mathcal{B}_{\mathrm{bln}}(\frac{1}{c_l}) \cap \mathcal{B}_\mu$ was proved at the beginning, $(U_{t+1}, V_{t+1})$ satisfy eq. (27), which concludes the proof. $\qquad\square$

### C.3 Auxiliary Lemmas for Theorem 3.2

The first result is Lemma SM2.7 in (Zilber and Nadler, 2022).

**Lemma C.8.** *Let* $Z = \binom{U}{V}, Z' = \binom{U'}{V'} \in \mathbb{R}^{(n_1+n_2) \times r}$. *Denote*

$$a = \left( \sqrt{2} \max\{\sigma_1(U), \sigma_1(V)\} d_P(Z', Z) + \tfrac{1}{2} d_P(Z', Z) \right) d_P(Z', Z). \tag{39}$$

*Then*

$$\|U'^\top U' - V'^\top V'\|_F \leq \|U^\top U - V^\top V\|_F + 2a, \tag{40}$$

$$\|U'V'^\top - UV^\top\|_F \leq a. \tag{41}$$

The second lemma is proven by Yi et al. (2016) as part of their proof of Theorem 3. It bounds the spectral distance of an initial estimate of $L^*$ obtained by removing the largest-magnitude entries and projecting to rank-$r$ matrices.

**Lemma C.9.** *Let* $X = L^* + S^*$ *where* $L^* \in \mathcal{M}(n_1, n_2, r, \mu)$ *with singular values* $\sigma_1^* \geq \cdots \geq \sigma_r^*$, *and* $S^*$ *satisfies Assumption 3 for some known* $0 < \alpha < 1$. *Let* $L = UV^\top$ *be an estimate of* $L^*$ *constructed according to the following equation,*

$$(U, V) = \mathrm{b\text{-}SVD}_r \left[ \tfrac{1}{p} \left( \mathcal{P}_\Omega(X) - \mathcal{T}_\alpha \left( \mathcal{P}_\Omega(X), \Omega \right) \right) \right]. \tag{42}$$

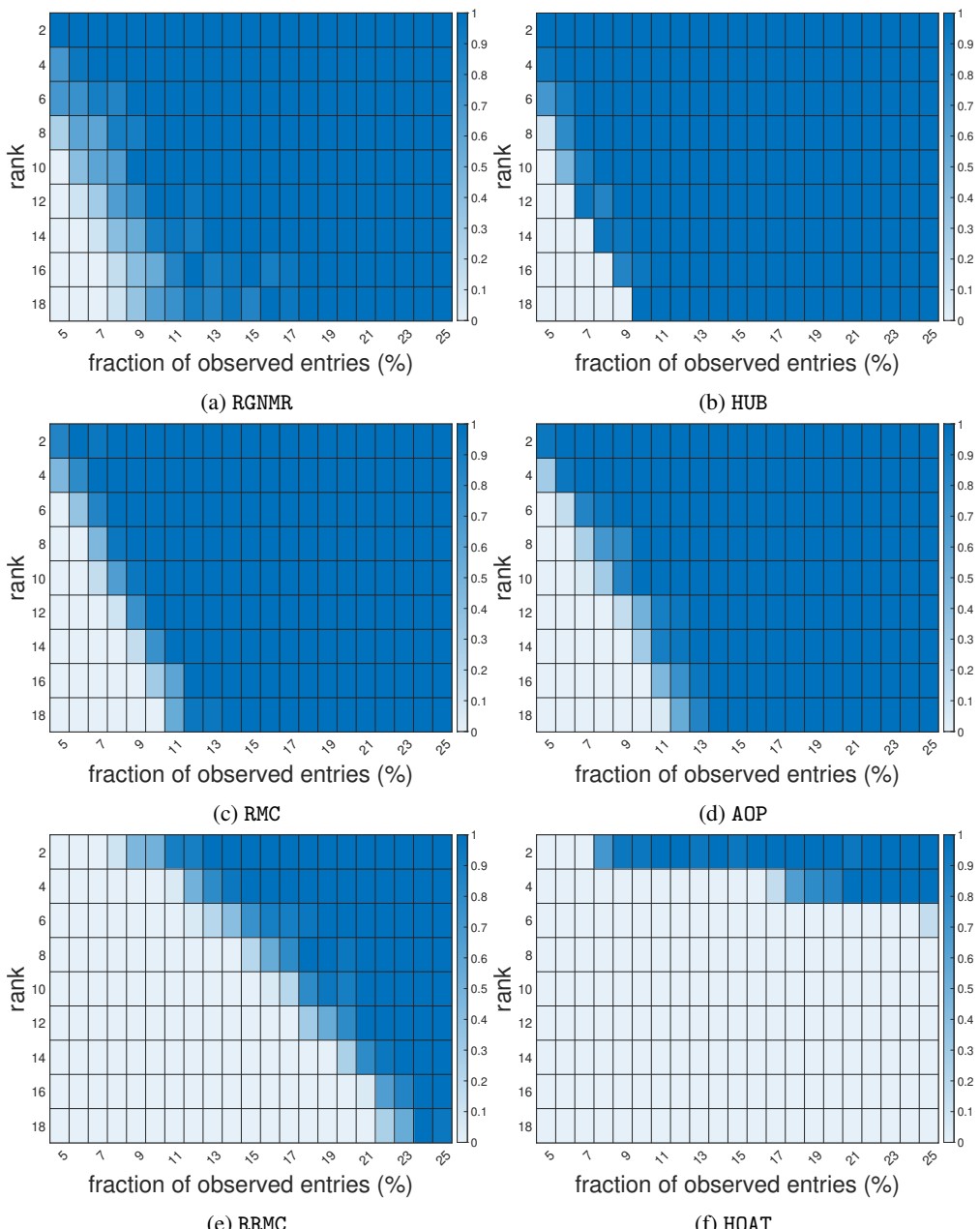

Figure 12: Performance of RMC methods as a function of the rank and the fraction of observed entries $\frac{|\Omega|}{n_1 n_2}$. Deep blue corresponds to $0\%$ failure, white corresponds to $100\%$ failure. The matrix is of size $1000 \times 1000$, the fraction of corrupted entries is $\alpha = 10\%$ and the condition number is $\kappa = 2$. Each point corresponds to 20 independent realizations.

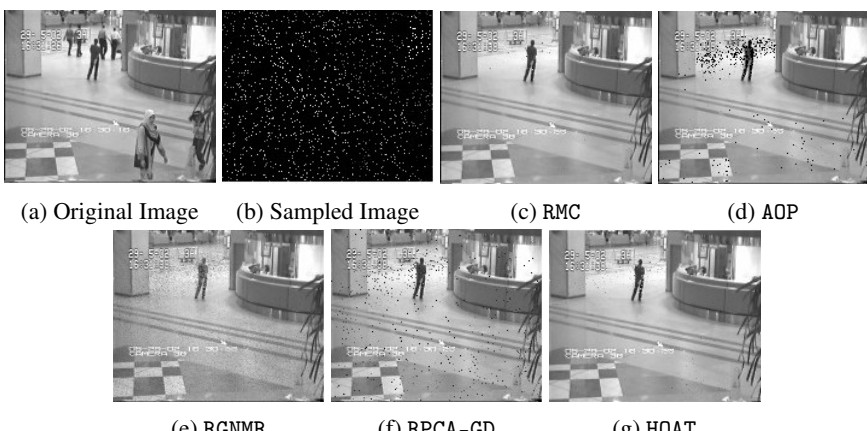

(a) Original Image    (b) Sampled Image    (c) RMC    (d) AOP

(e) RGNMR    (f) RPCA-GD    (g) HOAT

Figure 13: Background extraction for "Hall" video data. The frames are recovered from $5\%$ of the original entries with an input rank of $r = 1$.

*There exist a constant $c$ such that if $p \geq 4\frac{\mu r^2 \log n_1}{\epsilon^2 n_2}$, then w.p. at least $1 - \frac{6}{n_1}$ the spectral distance between $L$ and $L^*$ is bounded as follows,*

$$\|L - L^*\|_{op} \leq 16\alpha\mu r\sigma_1^* + \frac{2c\epsilon\sigma_1^*}{\sqrt{r}}. \tag{43}$$

The third lemma is a direct consequence of Lemma SM2.4 in (Zilber and Nadler, 2022).

**Lemma C.10.** *Let $L^*$ be a matrix of rank $r$ and let $Z^* = $ b-SVD$_r[L^*]$. For any $Z = \begin{pmatrix} U \\ V \end{pmatrix}$ that is perfectly balanced, namely $\|U^\top U - V^\top V\|_F = 0$, it holds that*

$$d_P(Z, Z^*) \leq \frac{\sqrt{2r}}{\sqrt{(\sqrt{2}-1)\sigma_r^*}}\|UV^\top - L^*\|_{op}. \tag{44}$$

Recall the definition of $S_{init}$, eq. (16). The following lemma bounds the operator norm of $Z = $ b-SVD$_r\left[\frac{1}{p}\left(\mathcal{P}_\Omega(X) - S_{init}\right)\right]$. Its proof appears in Appendix D.4.

**Lemma C.11.** *Let $L^* \in \mathcal{M}(n_1, n_2, r, \mu)$ with largest singular value $\sigma_1^*$. Let $Z^* = \begin{pmatrix} U^* \\ V^* \end{pmatrix} = $ b-SVD$_r[L^*]$ and $Z = \begin{pmatrix} U \\ V \end{pmatrix}$ where $U, V$ are perfectly balanced. If $\|UV^\top - L^*\|_{op} \leq \frac{\sigma_1^*}{4}$ then*

$$\|Z\|_{op} = \sqrt{2\sigma_1(UV^\top)} \leq \sqrt{\frac{5}{2}\sigma_1^*} \tag{45}$$

In the last step of the algorithm we apply the clipping operator $R_\eta$, see eq. (18). The following lemma , whose proof appears in Appendix D.5, state that under some conditions applying the clipping operator with the right $\eta$ reduces the Procrustes distance.

**Lemma C.12.** *Let $L^* \in \mathcal{M}(n_1, n_2, r, \mu)$, $Z^* = \begin{pmatrix} U^* \\ V^* \end{pmatrix} = $ b-SVD$_r[L^*]$ and $Z = \begin{pmatrix} U \\ V \end{pmatrix} = $ b-SVD$_r[A]$ for some matrix $A \in \mathbb{R}^{n_1 \times n_2}$. If $\|UV^\top - L^*\|_{op} \leq \frac{\sigma_1^*}{4}$ then for $\eta_1 = \sqrt{\frac{\mu r}{n_1}}\|Z\|_{op}$, $\eta_2 = \sqrt{\frac{\mu r}{n_2}}\|Z\|_{op}$ and $Z' = \begin{pmatrix} R_{\eta_1}(U) \\ R_{\eta_2}(V) \end{pmatrix}$ it follows that*

$$d_P(Z', Z^*) \leq d_P(Z, Z^*). \tag{46}$$

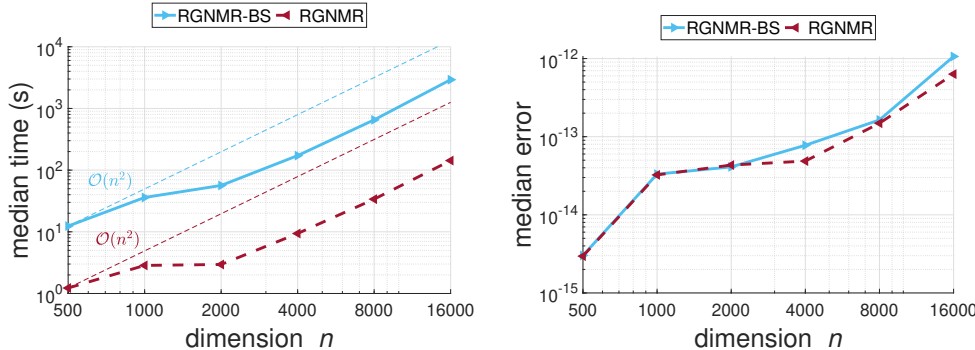

Figure 14: Effect of the matrix size on `RGNMR` performance. Both $x$ and $y$ axes are on a log scale. (left) Median run time of `RGNMR` and `RGNMR -BS` for a matrix of size $n \times n$ as a function of $n$. The dashed lines have a slope of 2. (right) Median Error as a function of the matrix size. The matrix has rank $r = 5$ and a condition number $\kappa = 2$. The oversampling ratio is $\rho = \frac{|\Omega|}{r(2n-r)} = 6$ and the fraction of corruption is $\alpha = 5\%$. Each point corresponds to 20 independent realizations.

## C.4 Proof of Theorem 3.2

*Proof.* First we would prove that $L_0 = U_0 V_0^\top$ is a sufficiently accurate estimate of $L^*$ and that it is balanced. Formally we would like to bound the terms $\|U_0^\top U_0 - V_0^\top V_0\|_F$ and $\|L_0 - L^*\|_F$. We denote $Z^* = \begin{pmatrix} U^* \\ V^* \end{pmatrix} = \text{b-SVD}_r[L^*]$ and recall that $Z_0 = \begin{pmatrix} U_0 \\ V_0 \end{pmatrix}$. We denote

$$a = \left(\sqrt{2\sigma_1^*} + \frac{1}{2}d_P(Z_0, Z^*)\right)d_P(Z_0, Z^*).$$

Since $\|U^{*\top}U^* - V^{*\top}V^*\| = 0$ and $U^*V^{*\top} = L^*$ by eq. (40) and eq. (41) of Lemma C.8

$$\begin{aligned} \|U_0^\top U_0 - V_0^\top V_0\|_F &\le 2a, \\ \|L_0 - L^*\|_F &\le a. \end{aligned} \tag{47}$$

To bound $a$ we need to bound $d_P(Z_0, Z^*)$. We do so by showing that by Lemma C.12 $d_P(Z_0, Z^*) \le d_P(Z, Z^*)$. To apply the lemma we first prove its conditions hold. Specifically we prove that $\|UV^\top - L^*\|_{op} \le \frac{\sigma_1^*}{4}$. Recall that $S_{init} = \mathcal{T}_\alpha(\mathcal{P}_\Omega(X), \Omega)$ and that $Z = \begin{pmatrix} U \\ V \end{pmatrix} = \text{b-SVD}_r\left[\frac{1}{p}\left(\mathcal{P}_\Omega(X) - S_{init}\right)\right]$. Since $\Omega$ follows Assumption 2 with $p \ge C\frac{\mu r^2 \kappa^4 \log n_1}{n_2}$, for a large enough $C$, we can apply Lemma C.9 with $\epsilon = \frac{1}{16c}$. By Lemma C.9 w.p. at least $1 - \frac{6}{n_1}$

$$\|UV^\top - L^*\|_{op} \le 16\alpha\mu r\sigma_1^* + \frac{2c\epsilon\sigma_1^*}{\sqrt{r}} \le 16\alpha\mu r\sigma_1^* + \frac{\sigma_1^*}{8\sqrt{r}}.$$

Since we assumed $\alpha \le \frac{1}{c_\alpha \kappa^2 r^{\frac{3}{2}} \mu}$, for a large enough $c_\alpha$

$$\|UV^\top - L^*\|_{op} \le \frac{16\sigma_1^*}{c_\alpha \kappa^2 \sqrt{r}} + \frac{\sigma_1^*}{8\sqrt{r}} \le \frac{\sigma_1^*}{4}.$$

Since $\eta_1 = \sqrt{\frac{\mu r}{n_1}}\|Z\|_{op}$, $\eta_2 = \sqrt{\frac{\mu r}{n_2}}\|Z\|_{op}$ and $Z_0 = \begin{pmatrix} R_{\eta_1}(U) \\ R_{\eta_2}(V) \end{pmatrix}$ by Lemma C.12

$$d_P(Z_0, Z^*) \le d_P(Z, Z^*).$$

Hence,

$$a \le \left(\sqrt{2\sigma_1^*} + \frac{1}{2}d_P(Z, Z^*)\right)d_P(Z, Z^*). \tag{48}$$

Next we explicitly bound $d_P(Z, Z^*)$. Since $Z$ is obtained using b-$\text{SVD}_r$, $\|U^\top U - V^\top V\|_F = 0$. Therefore by Lemma C.10

$$d_P(Z, Z^*) \leq \frac{\sqrt{2r}}{\sqrt{(\sqrt{2} - 1)\sigma_r^*}} \|UV^\top - L^*\|_{op}. \tag{49}$$

To bound $\|UV^\top - L^*\|_{op}$ we apply Lemma C.9. Since $\Omega$ follows Assumption 2 with $p \geq C\frac{\mu r^2 \kappa^4 \log n_1}{n_2}$ we can apply Lemma C.9 with $\epsilon = \frac{2}{\sqrt{C}k^2}$. This gives that w.p. at least $1 - \frac{6}{n_1^2}$

$$d_P(Z, Z^*) \leq \frac{\sqrt{2r}}{\sqrt{(\sqrt{2} - 1)\sigma_r^*}} \left( 16\alpha\mu r\sigma_1^* + \frac{2c\epsilon\sigma_1^*}{\sqrt{r}} \right) = 3\sqrt{\sigma_r^*} \left( 16\alpha\mu\kappa r^{\frac{3}{2}} + \frac{4c}{\sqrt{C}\kappa} \right).$$

Since we assumed $\alpha \leq \frac{1}{c_\alpha \kappa^2 r^{\frac{3}{2}} \mu}$

$$d_P(Z, Z^*) \leq 3\sqrt{\sigma_r^*} \left( \frac{16}{c_\alpha \kappa} + \frac{4c}{\sqrt{C}\kappa} \right) \leq 3\frac{\sqrt{\sigma_r^*}}{\kappa} \left( \frac{16}{c_\alpha} + \frac{4c}{\sqrt{C}} \right). \tag{50}$$

Denote $\xi = \left( \frac{16}{c_\alpha} + \frac{4c}{\sqrt{C}} \right)$. Inserting eq. (50) into eq. (48) gives

$$a \leq \left( \sqrt{2\sigma_1^*} + 3\frac{\sqrt{\sigma_r^*}}{\kappa}\xi \right) 3\frac{\sqrt{\sigma_r^*}}{\kappa}\xi = \frac{\sigma_r^*}{\sqrt{\kappa}} \left( 3\sqrt{2} + \frac{9}{\kappa^{1.5}}\xi \right) \xi.$$

Note that for any $\varepsilon > 0$ for large enough $C, c_\alpha$ it follows that $\xi < \varepsilon$. Therefore for large enough constants $c_\alpha$ and $C$

$$a \leq \min \left\{ \frac{\sigma_r^*}{c_e\sqrt{\kappa}}, \frac{\sigma_r^*}{4c_l} \right\}.$$

Inserting the above bound into eq. (47) gives

$$(U_0, V_0) \in \mathcal{B}_{\text{err}}(\frac{1}{c_e\sqrt{\kappa}}) \cap \mathcal{B}_{\text{bln}}(\frac{1}{2c_l}).$$

Next we prove that $(U_0, V_0) \in \mathcal{B}_\mu$, which requires to bound $\|U_0\|_{2,\infty}$ and $\|V_0\|_{2,\infty}$. Since $\begin{pmatrix} U_0 \\ V_0 \end{pmatrix} = \begin{pmatrix} R_{\eta_1}(U) \\ R_{\eta_2}(V) \end{pmatrix}$ by the definition, eq. (18), of the clipping operator $R_\eta$

$$\|U_0\|_{2,\infty} \leq \eta_1 = \sqrt{\frac{\mu r}{n}}\|Z\|_{op}, \quad \|V_0\|_{2,\infty} \leq \eta_2 = \sqrt{\frac{\mu r}{n_2}}\|Z\|_{op}.$$

Since we proved that $\|UV^\top - L^*\| \leq \frac{\sigma_1^*}{4}$ we can apply Lemma C.11. Specifically inserting eq. (45) into the above inequality completes the proof as it gives

$$\|U_0\|_{2,\infty} \leq \sqrt{\frac{\mu r}{n_1}}\|Z\|_{op} = \sqrt{\frac{2\mu r\sigma_1(UV^\top)}{n_1}} \leq \sqrt{\frac{5\sigma_1^*}{2}\frac{\mu r}{n_1}} \leq \sqrt{\frac{3\mu r\sigma_1^*}{n_1}},$$

$$\|V_0\|_{2,\infty} \leq \sqrt{\frac{\mu r}{n_2}}\|Z\|_{op} = \sqrt{\frac{2\mu r\sigma_1(UV^\top)}{n_2}} \leq \sqrt{\frac{5\sigma_1^*}{2}\frac{\mu r}{n_2}} \leq \sqrt{\frac{3\mu r\sigma_1^*}{n_2}}.$$

$\square$

# D   Proofs of Lemmas C.4, C.5, C.6, C.11 and C.12.

## D.1   Proof of Lemma C.4

To prove the lemma we use the following auxiliary result, whose proof appears in Appendix E.1.

**Lemma D.1.** *Let $Z_t$ be the estimated factorization after $t$ steps as defined in Algorithm 2, and let $Z_{t+1}$ be the factorization at iteration $t + 1$. Denote*

$$a_t = \sqrt{2} \max\{\sigma_1(U_t), \sigma_1(V_t)\} d_P(Z_{t+1}, Z_t) + \tfrac{1}{2} d_P^2(Z_{t+1}, Z_t).$$

*For large enough constants $c_e, c_l$ the following holds. If $(U_0, V_0) \in \mathcal{B}_{\mathrm{err}}(1/c_e\sqrt{\kappa}) \cap \mathcal{B}_{\mathrm{bln}}(1/2c_l) \cap \mathcal{B}_\mu$ and $(U_s, V_s) \in \mathcal{C}(U_{s-1}, V_{s-1}, \frac{\sigma_r^*}{4^s c_e \kappa})$ for every $1 \le s \le t + 1$ then*

$$a_t \le \frac{4\sqrt{2}\sigma_r^*}{2^{t+1}\sqrt{c_e}} + \frac{\sigma_r^*}{2 \cdot 4^{t+1} c_e \kappa}. \tag{51}$$

*Proof of Lemma C.4.* We start by proving that $(U_{t+1}, V_{t+1}) \in \mathcal{B}_{\mathrm{bln}}(\frac{1}{c_l})$. By eq. (40) of Lemma C.8,

$$\|U_{t+1}^\top U_{t+1} - V_{t+1}^\top V_{t+1}\| \le \|U_t^\top U_t - V_t^\top V_t\| + 2a_t.$$

Therefore, by induction

$$\|U_{t+1}^\top U_{t+1} - V_{t+1}^\top V_{t+1}\| \le \|U_0^\top U_0 - V_0^\top V_0\| + 2\sum_{k=0}^t a_k. \tag{52}$$

By eq. (51) of Lemma D.1, for large enough $c_e$, for each iteration step $0 \le k \le t$

$$a_k \le \frac{\sigma_r^*}{2^{k+3} c_l}. \tag{53}$$

Inserting eq. (53) into eq. (52) gives,

$$\|U_{t+1}^\top U_{t+1} - V_{t+1}^\top V_{t+1}\| \le \|U_0^\top U_0 - V_0^\top V_0\| + \frac{\sigma_r^*}{c_l} \sum_{k=0}^t \frac{1}{2^{k+2}}$$

$$\le \|U_0^\top U_0 - V_0^\top V_0\| + \frac{\sigma_r^*}{2c_l}.$$

Since $(U_0, V_0) \in \mathcal{B}_{\mathrm{bln}}(1/2c_l)$ the above inequality implies that $(U_{t+1}, V_{t+1}) \in \mathcal{B}_{\mathrm{bln}}(1/c_l)$.

Next, we prove $(U_{t+1}, V_{t+1}) \in \mathcal{B}_{\mathrm{err}}(\frac{13}{\sqrt{c_e}})$. By the triangle inequality

$$\|U_{t+1}V_{t+1}^\top - L^*\|_F \le \|U_{t+1}V_{t+1}^\top - U_t V_t^\top\|_F + \|U_t V_t^\top - L^*\|_F.$$

By Lemma C.8 $\|U_{t+1}V_{t+1}^\top - U_t V_t^\top\|_F \le a_t$ and therefore,

$$\|U_{t+1}V_{t+1}^\top - L^*\|_F \le a_t + \|U_t V_t^\top - L^*\|_F.$$

By induction it follows that

$$\|U_{t+1}V_{t+1}^\top - L^*\|_F \le \|U_0 V_0^\top - L^*\|_F + \sum_{k=0}^t a_k. \tag{54}$$

Next we use D.1 to bound $a_k$. Note that for $c_e \ge 1$ since $\kappa \ge 1$,

$$a_k \le \frac{4\sqrt{2}\sigma_r^*}{2^{k+1}\sqrt{c_e}} + \frac{\sigma_r^*}{2 \cdot 4^{k+1}\sqrt{c_e}} \le \frac{4\sqrt{2}\sigma_r^*}{2^{k+1}\sqrt{c_e}} + \frac{4\sqrt{2}\sigma_r^*}{2^{k+1}\sqrt{c_e}} \le \frac{4\sqrt{2}\sigma_r^*}{2^k\sqrt{c_e}}.$$

As a result of the above inequality

$$\sum_{k=0}^t a_k \le \frac{4\sqrt{2}\sigma_r^*}{\sqrt{c_e}} \sum_{k=0}^t \frac{1}{2^k} \le \frac{8\sqrt{2}\sigma_r^*}{\sqrt{c_e}}. \tag{55}$$

By the assumption $(U_0, V_0) \in \mathcal{B}_{\mathrm{err}}\left(\frac{1}{c_e\sqrt{\kappa}}\right)$ it follows that $\|U_0 V_0^\top - L^*\| \le \frac{\sigma_r^*}{c_e\sqrt{\kappa}}$. Inserting this inequality and eq. (55) into eq. (54) gives

$$\|U_{t+1}V_{t+1}^\top - L^*\|_F \le \frac{\sigma_r^*}{c_e\sqrt{\kappa}} + \frac{(8\sqrt{2})\sigma_r^*}{\sqrt{c_e}} \le \frac{13\sigma_r^*}{\sqrt{c_e}}.$$

We conclude that $(U_{t+1}, V_{t+1}) \in \mathcal{B}_{\mathrm{err}}(\frac{13}{\sqrt{c_e}}) \cap \mathcal{B}_{\mathrm{bln}}(\frac{1}{c_l})$. $\qquad\square$

## D.2 Proof of Lemma C.5

To prove the Lemma C.5 we use the following four auxiliary lemmas, whose proofs are in Appendix E.2. All of them consider the following setting. Let $L_t = U_t V_t^\top$ and $\Lambda_t$ be the estimates of $L^*$ and $\Lambda_*$ at iteration $t$, respectively. Let $L_{t+1} = U_{t+1} V_{t+1}^\top$ be the updated estimate, where $(U_{t+1}, V_{t+1})$ are computed by (10) with some $\delta > 0$. Denote $\Delta U_t^* = U_t - U^*$, $\Delta U_{t+1} = U_{t+1} - U_t$ with similar definitions for $\Delta V_t^*$ and $\Delta V_{t+1}$. Denote by $\Omega_t = \Omega \setminus \Lambda_t$ the set of non removed entries, by $I_t^{in} = \Omega_t \cap \Lambda_*^c$ the set of non-removed entries which are true inliers and by $I_t^{out} = \Omega_t \cap \Lambda_*$ the set of non-removed entries which are outliers. Finally, recall that for factor matrices $(U, V)$ and a subset of entries $\Lambda$ the objective function value is

$$\mathcal{L}_\Lambda^t(U, V) = \arg\min_{U,V} \|U_t V^\top + U V_t^\top - U_t V_t^\top - X\|_{F(\Lambda)}.$$

**Lemma D.2.** *The error restricted to the set $I_t^{in}$ is bounded as follows,*

$$\|L_{t+1} - L^*\|_{F(I_t^{in})} \le \mathcal{L}_{I_t^{in}}^t(U_{t+1}, V_{t+1}) + \|\Delta U_{t+1} \Delta V_{t+1}^\top\|_{F(I_t^{in})}. \tag{56}$$

To upper bound the first term on the RHS of eq. (56) we use the following lemma.

**Lemma D.3.** *Assume that $L_t$ is sufficiently close to $L^*$ in the sense that the set $\mathcal{A}_t = \mathcal{B}^* \cap \mathcal{B}_\mu \cap \mathcal{C}\left(U_t, V_t, \frac{\delta}{4^{t+1}}\right)$ is non empty. Then*

$$\mathcal{L}_{I_t^{in}}^t(U_{t+1}, V_{t+1}) \le \sqrt{2} \mathcal{L}_{\Omega_t}^t(U^*, V^*) - \mathcal{L}_{I_t^{out}}^t(U_{t+1}, V_{t+1}), \tag{57}$$

*for any $(U^*, V^*) \in \mathcal{A}_t$, an exact factorization of $L^*$.*

To upper bound the RHS of eq. (57) we employ the following two lemmas. The first lemma upper bounds the first term and the second lemma lower bounds the second term.

**Lemma D.4.** *For any $(U^*, V^*) \in \mathcal{B}^*$*

$$\mathcal{L}_{\Omega_t}^t(U^*, V^*) \le \|\Delta U_t^* \Delta V_t^{*\top}\|_{F(\Omega_t)} + \|S^*\|_{F(\Omega_t)}. \tag{58}$$

**Lemma D.5.** *Under the above assumptions,*

$$\mathcal{L}_{I_t^{out}}^t(U_{t+1}, V_{t+1}) \ge \|L_{t+1} - L^* - S^*\|_{F(I_t^{out})} - \|\Delta U_{t+1} \Delta V_{t+1}^\top\|_{F(I_t^{out})}. \tag{59}$$

*Proof of Lemma C.5.* Our goal is to bound $\|L_{t+1} - L^*\|_{F(\Omega_t)}$. Since $\Omega_t = I_t^{in} \cup I_t^{out}$,

$$\|L_{t+1} - L^*\|_{F(\Omega_t)} \le \|L_{t+1} - L^*\|_{F(I_t^{in})} + \|L_{t+1} - L^*\|_{F(I_t^{out})}.$$

Lemma D.2 bounds $\|L_{t+1} - L^*\|_{F(I_t^{in})}$,

$$\|L_{t+1} - L^*\|_{F(\Omega_t)} \le \mathcal{L}_{I_t^{in}}^t(U_{t+1}, V_{t+1}) + \|\Delta U_{t+1} \Delta V_{t+1}^\top\|_{F(I_t^{in})} + \|L_{t+1} - L^*\|_{F(I_t^{out})}. \tag{60}$$

To upper bound $\mathcal{L}_{I_t^{in}}^t(U_{t+1}, V_{t+1})$ we insert eq. (59) and eq. (58) into eq. (57) which yields,

$$\begin{aligned}
\mathcal{L}_{I_t^{in}}^t(U_{t+1}, V_{t+1}) \le & \sqrt{2} \|\Delta U_t^* \Delta V_t^{*\top}\|_{F(\Omega_t)} + \sqrt{2} \|S^*\|_{F(\Omega_t)} \\
& + \|\Delta U_{t+1} \Delta V_{t+1}^\top\|_{F(I_t^{out})} - \|L_{t+1} - L^* - S^*\|_{F(I_t^{out})}.
\end{aligned} \tag{61}$$

Inserting eq. (61) into eq. (60) and using the fact that $\|S\|_{F(\Omega_t)} = \|S^*\|_{F(I_t^{out})}$ gives,

$$\begin{aligned}
\|L_{t+1} - L^*\|_{F(\Omega_t)} \le & \sqrt{2} \|\Delta U_t^* \Delta V_t^{*\top}\|_{F(\Omega_t)} + \sqrt{2} \|S^*\|_{F(I_t^{out})} \\
& + \underbrace{\|\Delta U_{t+1} \Delta V_{t+1}^\top\|_{F(I_t^{out})} + \|\Delta U_{t+1} \Delta V_{t+1}^\top\|_{F(I_t^{in})}}_{\Delta} \\
& + \|L_{t+1} - L^*\|_{F(I_t^{out})} - \|L_{t+1} - L^* - S^*\|_{F(I_t^{out})}.
\end{aligned} \tag{62}$$

To bound the term $\Delta$ above we note that,

$$\|\Delta U_{t+1} \Delta V_{t+1}^\top\|_{F(I_t^{out})}^2 + \|\Delta U_{t+1} \Delta V_{t+1}^\top\|_{F(I_t^{in})}^2 = \|\Delta U_{t+1} \Delta V_{t+1}^\top\|_{F(\Omega_t)}^2.$$

Using the inequality $a + b \leq \sqrt{2}\sqrt{a^2 + b^2}$ yields

$$\Delta \leq \sqrt{2}\|\Delta U_{t+1}\Delta V_{t+1}^\top\|_{F(\Omega_t)}. \tag{63}$$

By the triangle inequality

$$\|L_{t+1} - L^*\|_{F(I_t^{\text{out}})} - \|L_{t+1} - L^* - S^*\|_{F(I_t^{\text{out}})} \leq \|S^*\|_{F(I_t^{\text{out}})}. \tag{64}$$

Inserting eq. (63) and eq. (64) into eq. (62) gives,

$$\begin{aligned}
\|L_{t+1} - L^*\|_{F(\Omega_t)} \leq &\sqrt{2}\left[\|\Delta U_t^*\Delta V_t^{*\top}\|_{F(\Omega_t)} + \|\Delta U_{t+1}\Delta V_{t+1}^\top\|_{F(\Omega_t)}\right] \\
&+ (1 + \sqrt{2})\|S^*\|_{F(I_t^{\text{out}})}.
\end{aligned}$$

$\square$

## D.3 Proofs of Lemmas C.6 and C.7

Before we prove the lemmas we present three auxiliary lemmas. The first is Lemma 10 in (Yi et al., 2016). It establishes that if the sampling probability $p$ is sufficiently high, then the number of sampled entries in all rows and columns of $\Omega$ concentrates around their expected value.

**Lemma D.6.** *Let $\Omega \subset [n_1] \times [n_2]$ with $n_1 \geq n_2$ satisfy Assumption 1, namely its entries are sampled independently with probability $p$. There exist a constant $C$ such that if $p \geq C\frac{\log n_1}{n_2}$ then w.p. at least $1 - \frac{6}{n_1}$ uniformally over all $i \in [n_1]$ and $j, \in [n_2]$*

$$|r_i - pn_2| \leq \frac{1}{2}pn_2, \quad |c_j - pn_1| \leq \frac{1}{2}pn_1. \tag{65}$$

The second result is Lemma 14 in (Yi et al., 2016) . It provides an upper bound on the error $\|UV^\top - U^*V^{*\top}\|_{F(\Lambda)}$ for a sufficiently well spread out subset $\Lambda$ of the entries.

**Lemma D.7.** *Let $L^*$ have rank $r$, largest singular value $\sigma_1^*$ and decomposition $(U^*, V^*) \in \mathcal{B}_* \cap \mathcal{B}_\mu$. Let $U \in \mathbb{R}^{n_1 \times r}, V \in \mathbb{R}^{n_2 \times r}$ be a pair of matrices, such that $(U, V) \in \mathcal{B}_\mu$, and let $\theta \in (0, 1)$. Assume that $\Lambda \subseteq [n_1] \times [n_2]$ satisfies $|\Lambda_{(i,\cdot)}| \leq \theta n_2$ for all $i \in [n_1]$ and $|\Lambda_{(\cdot,j)}| \leq \theta n_1$ for all $j \in [n_2]$ then*

$$\|UV^\top - U^*V^{*\top}\|_{F(\Lambda)}^2 \leq 18\sqrt{3}\theta r\mu\sigma_1^*(\|U - U^*\|_F^2 + \|V - V^*\|_F^2). \tag{66}$$

The third lemma bounds the impact of the entries in $(\Omega \setminus \Lambda_t) \cap \Lambda_*$ on the error term $\|L_t - X\|_{F(\Omega)}$. The lemma is based on a work presented in (Yi et al., 2016) (as part of their proof of Lemma 2), its proof appears in Appendix E.3.

**Lemma D.8.** *Let $X = L^* + S^*$, and $\Omega \subset [n_1] \times [n_2]$, where the corruption matrix $S^*$ satisfies Assumption 3 for some known $0 < \alpha < 1$. Suppose that for a given estimate $L$ of $L^*$, the set of corrupted entries is estimated by $\Lambda = \text{support}(\mathcal{T}_{\gamma\alpha}(L - X, \Omega))$ with an over remvoal factor $1 < \gamma \leq \frac{1}{\alpha}$. Then the error on the remaining corrupted entries $(\Omega \setminus \Lambda) \cap \Lambda^*$, where $\Lambda^* = \text{support}(S^*)$, is bounded as follows,*

$$\|L - X\|_{F((\Omega \setminus \Lambda) \cap \Lambda_*)} \leq \sqrt{\frac{2}{\gamma - 1}}\|L - L^*\|_{F(\Omega)}. \tag{67}$$

*Proof of Lemma C.6.* By the triangle inequality

$$\begin{aligned}
\|S^*\|_{F((\Omega \setminus \Lambda) \cap \Lambda_*)} &= \|L - (L^* + S^*) - (L - L^*)\|_{F((\Omega \setminus \Lambda) \cap \Lambda_*)} \\
&\leq \underbrace{\|L - L^*\|_{F((\Omega \setminus \Lambda) \cap \Lambda_*)}}_{W_1} + \underbrace{\|(L - L^*) - S^*\|_{F((\Omega \setminus \Lambda) \cap \Lambda_*)}}_{W_2}.
\end{aligned} \tag{68}$$

First we bound $W_1$. To this end we note that since $\Omega$ follows Assumption 2 with $p \geq C\frac{\log n_1}{n_2}$ by Lemma D.6 w.p. at least $1 - \frac{6}{n_1}$ for every $i \in [n_1]$ and $j \in [n_2]$

$$|r_i - pn_2| \leq \frac{1}{2}pn_2, \quad |c_j - pn_1| \leq \frac{1}{2}pn_1.$$

By Assumption 3, $S^* \in \mathcal{S}_\alpha^\Omega$ and therefore

$$|(\Lambda_*)_{(i,\cdot)}| \leq \alpha r_i, \quad |(\Lambda_*)_{(\cdot,j)}| \leq \alpha c_j,$$

Since $(\Omega \setminus \Lambda_t) \cap \Lambda_* \subset \Lambda_*$

$$|((\Omega \setminus \Lambda_t) \cap \Lambda_*)_{(i,\cdot)}| \leq |(\Lambda_*)_{(i,\cdot)}| \leq \alpha r_i \leq \frac{3}{2}\alpha p n_2$$

and similarly $|(\Lambda_*)_{(\cdot,j)}| \leq \frac{3}{2}\alpha p n_1$. Therefore we can invoke Lemma D.7 with $\theta = \frac{3}{2}\alpha p$ to obtain the following bound on $W_1$,

$$W_1^2 = \|UV^\top - U^*V^{*\top}\|^2_{F((\Omega\setminus\Lambda)\cap\Lambda_*)} \leq 27\sqrt{3}\alpha pr\mu\sigma_1^*(\|U - U^*\|_F^2 + \|V - V^*\|_F^2).$$

Since $(U^*, V^*)$ satisfy eq. (24) it follows that

$$W_1 \leq \sqrt{27\sqrt{3}\alpha pr\mu\sigma_1^* \frac{25}{4\sigma_r^*}\|L - L\|_F^2} \leq 18\sqrt{\alpha p\mu r\kappa}\|L - L^*\|_F.$$

Next we bound $W_2$ in eq. (68). To this end we invoke Lemma D.8 which gives,

$$W_2 = \|L_t - L^* - S^*\|_{F((\Omega\setminus\Lambda_t)\cap\Lambda_*)} \leq \sqrt{\frac{2}{\gamma - 1}}\|L_t - L^*\|_{F(\Omega)}.$$

Inserting the bounds on $W_1$ and $W_2$ into eq. (68) yields eq. (25). $\qquad\square$

*Proof of Lemma C.7.* Since $\Omega$ follows Assumption 2 with $p \geq C\frac{\log n_1}{n_2}$ by Lemma D.6 w.p. at least $1 - \frac{6}{n_1}$ for every $i \in [n_1]$ and $j \in [n_2]$

$$|r_i - pn_2| \leq \frac{1}{2}pn_2, \quad |c_j - pn_1| \leq \frac{1}{2}pn_1.$$

By the definition of $\mathcal{T}_{\gamma\alpha}$, eq. (6), for any matrix $A$, it follows that if $\Lambda = \text{support}(\mathcal{T}_{\gamma\alpha}(A, \Omega))$ then

$$|(\Lambda)_{(i,\cdot)}| \leq \gamma\alpha r_i \leq \frac{3}{2}\gamma\alpha pn_2, \quad |(\Lambda)_{(\cdot,j)}| \leq \gamma\alpha c_j \leq \frac{3}{2}\gamma\alpha pn_1.$$

Using Lemma D.7 with $\Lambda$ and $\theta = \frac{3}{2}\gamma\alpha p$ we obtain eq. (26). $\qquad\square$

### D.4 Proof of Lemma C.11

*Proof.* First we bound $\sigma_1(UV^\top)$. By the assumption $\|UV^\top - L^*\|_{op} \leq \frac{\sigma_1^*}{4}$ and the triangle inequality,

$$\sigma_1(UV^\top) \leq \frac{5}{4}\sigma_1^*.$$

Next we bound $\|Z\|_{op}$. Since $U, V$ are perfectly balanced, there exists $\tilde{U} \in \mathbb{R}^{n_1 \times r}, \tilde{V} \in \mathbb{R}^{n_2 \times r}$ unitary matrices and $\tilde{\Sigma} \in \mathbb{R}^{r \times r}$ a diagonal matrix such that

$$U = \tilde{U}\tilde{\Sigma}^{\frac{1}{2}}, \quad \text{and} \quad V = \tilde{V}\tilde{\Sigma}^{\frac{1}{2}}.$$

Note that $UV^\top = \tilde{U}\tilde{\Sigma}\tilde{V}$, and thus $\sigma_1(UV^\top) = \lambda_1(\tilde{\Sigma})$. In addition, $Z^\top Z = U^\top U + V^\top V = 2\tilde{\Sigma}$. Hence,

$$\|Z\|_{op} = \sqrt{\lambda_1(Z^\top Z)} = \sqrt{\lambda_1(2\tilde{\Sigma})} = \sqrt{2\sigma_1(UV^\top)} \leq \sqrt{\frac{5\sigma_1^*}{2}}.$$

$\qquad\square$

### D.5 Proof of Lemma C.12

To prove the lemma we use the following auxiliary lemma which is Lemma 11 in (Zheng and Lafferty, 2016).

**Lemma D.9.** *Let $y \in \mathbb{R}^r$ be a vector such that $\|y\| \leq \eta$. Then for any $x \in \mathbb{R}^r$*

$$\|R_\eta(x) - y\|^2 \leq \|x - y\|^2. \tag{69}$$

*proof of Lemma C.12.* Recall that

$$d_p(Z', Z^*) = \min_P \|Z' - Z^* P\|_F, \quad d_p(Z, Z^*) = \min_P \|Z - Z^* P\|_F.$$

where $P$ is orthogonal. In what follows we prove that for any orthogonal matrix $\|Z' - Z^* P\|_F \leq \|Z - Z^* P\|_F$. Let $P$ be an orthogonal matrix,

$$\|Z' - Z^* P\|_F^2 = \sum_{i=1}^{n_1} \|R_{\eta_1}(U_{(i,\cdot)}) - (U^* P)_{(i,\cdot)}\|^2 + \sum_{i=n_1+1}^{n_1+n_2} \|R_{\eta_2}(V_{(i,\cdot)}) - (V^* P)_{(i,\cdot)}\|^2, \tag{70}$$

In order to bound $\|R_{\eta_1}(U_{(i,\cdot)}) - (U^* P)_{(i,\cdot)}\|^2$ and $\|R_{\eta_2}(V_{(i,\cdot)}) - (V^* P)_{(i,\cdot)}\|^2$ using Lemma D.9 we first need to prove that $\|(U^* P)_{(i,\cdot)}\| \leq \eta_1, \|(V^* P)_{(i,\cdot)}\| \leq \eta_2$. Since $L^*$ has an incoherence parameter of $\mu$

$$\|U^*\|_{2,\infty} \leq \sqrt{\frac{r\mu\sigma_1^*}{n_1}}, \quad \|V^*\|_{2,\infty} \leq \sqrt{\frac{r\mu\sigma_1^*}{n_2}}.$$

Since we assumed $\|UV^\top - L^*\|_{op} \leq \frac{\sigma_1^*}{4}$, by the triangle inequality $\sigma_1^* \leq 2\sigma_1(UV^\top)$. Therefore

$$\|U^*\|_{2,\infty} \leq \sqrt{\frac{r\mu\sigma_1^*}{n_1}} \leq \sqrt{\frac{2r\mu\sigma_1(UV^\top)}{n_1}},$$

$$\|V^*\|_{2,\infty} \leq \sqrt{\frac{r\mu\sigma_1^*}{n_2}} \leq \sqrt{\frac{2r\mu\sigma_1(UV^\top)}{n_2}}.$$

By Lemma C.11 $\sqrt{2\sigma_1(UV^\top)} = \|Z\|_{op}$, hence

$$\|U^*\|_{2,\infty} \leq \sqrt{\frac{r\mu}{n_1}} \|Z\|_{op} = \eta_1,$$

$$\|V^*\|_{2,\infty} \leq \sqrt{\frac{r\mu}{n_2}} \|Z\|_{op} = \eta_2.$$

Let $P$ be an orthogonal matrix then

$$\|U^* P\|_{2,\infty} \leq \|U^*\|_{2,\infty} \|P^\top\|_{op} = \|U^*\|_{2,\infty},$$
$$\|V^* P\|_{2,\infty} \leq \|V^*\|_{2,\infty} \|P^\top\|_{op} = \|V^*\|_{2,\infty}.$$

Therefore, $\|(U^* P)_{(i,\cdot)}\| \leq \eta_1, \|(V^* P)_{(i,\cdot)}\| \leq \eta_2$. By Lemma D.9

$$\|R_{\eta_1}(U_{(i,\cdot)}) - (U^* P)_{(i,\cdot)}\|^2 \leq \|U_{(i,\cdot)} - (U^* P)_{(i,\cdot)}\|^2,$$

and a similar result follows for $V^*$ with $\eta_2$. Inserting these bounds into eq. (70) gives

$$\|Z' - Z^* P\|_F^2 \leq \sum_{i=1}^{n_1} \|U_{(i,\cdot)} - (U^* P)_{(i,\cdot)}\|^2 + \sum_{i=n_1+1}^{n_1+n_2} \|V_{(i,\cdot)} - (V^* P)_{(i,\cdot)}\|^2 = \|Z - Z^* P\|_F^2.$$

Since this is true for any orthogonal matrix it follows that

$$d_P(Z', Z^*) \leq d_P(Z, Z^*).$$

$\square$

# E Proofs of Lemmas D.1, D.2, D.3, D.4, D.5,and D.8.

## E.1 Proof of Lemma D.1.

As part of the proof we use the following lemma, whose proof appears in Appendix F.

**Lemma E.1.** *There exist constants $c_e, c_l$ such that the following holds. If $(U_0, V_0) \in \mathcal{B}_{\mathrm{err}}\left(\frac{1}{c_e\sqrt{\kappa}}\right) \cap \mathcal{B}_{\mathrm{bln}}\left(\frac{1}{2c_l}\right) \cap \mathcal{B}_\mu$ then*

$$\max\{\sigma_1(U_0), \sigma_1(V_0)\} \leq 2\sqrt{\sigma_1^*}. \tag{71}$$

*Additionally if for every $1 \leq s \leq t + 1$, $(U_s, V_s) \in \mathcal{C}(U_{s-1}, V_{s-1}, \frac{\sigma_r^*}{4^s c_e \sqrt{\kappa}})$ then*

$$\max\{\sigma_1(U_{t+1}), \sigma_1(V_{t+1})\} \leq 4\sqrt{\sigma_1^*}. \tag{72}$$

*Proof of Lemma D.1.* Recall that

$$a_t = \sqrt{2}\max\{\sigma_1(U_t), \sigma_1(V_t)\}d_P(Z_{t+1}, Z_t) + \tfrac{1}{2}d_P^2(Z_{t+1}, Z_t).$$

Since for every $1 \leq s \leq t$, $(U_s, V_s) \in \mathcal{C}(U_{s-1}, V_{s-1}, \frac{\sigma_r^*}{4^s c_e \sqrt{\kappa}})$ by Lemma E.1 it follows that $\max\{\sigma_1(U_t), \sigma_1(V_t)\} \leq 4\sqrt{\sigma_1^*}$. Therefore

$$a_t \leq \left(\sqrt{32}\sqrt{\sigma_1^*} + \tfrac{1}{2}d_P(Z_t, Z_{t+1})\right)d_P(Z_t, Z_{t+1}).$$

Since $d_P(Z_t, Z_{t+1}) \leq \|Z_t - Z_{t+1}\|_F = \sqrt{\|U_{t+1} - U_t\|_F^2 + \|V_{t+1} - V_t\|_F^2}$ we can bound $a_t$ as follows,

$$a_t \leq \left(\sqrt{32\left(\|U_{t+1} - U_t\|_F^2 + \|V_{t+1} - V_t\|_F^2\right)}\sqrt{\sigma_1^*} + \tfrac{1}{2}\|U_{t+1} - U_t\|_F^2 + \|V_{t+1} - V_t\|_F^2\right).$$

Since $(U_{t+1}, V_{t+1}) \in \mathcal{C}\left(U_t, V_t, \frac{\sigma_r^*}{4^{t+1}c_e\kappa}\right)$, see eq. (8), $\|U_{t+1} - U_t\|_F^2 + \|V_{t+1} - V_t\|_F^2 \leq \frac{\sigma_r^*}{4^{t+1}c_e\kappa}$. Hence,

$$a_t \leq \frac{\sqrt{32\sigma_r^*\sigma_1^*}}{2^{t+1}\sqrt{\kappa c_e}} + \frac{\sigma_r^*}{2\cdot 4^{t+1}c_e\kappa} = \frac{4\sqrt{2}\sigma_r^*}{2^{t+1}\sqrt{c_e}} + \frac{\sigma_r^*}{2\cdot 4^{t+1}c_e\kappa}.$$

$\square$

## E.2 Proofs of Lemmas D.2, D.3, D.4 and D.5

*Proof of Lemma D.2.* We first lower bound $\mathcal{L}_{I_t^{\mathrm{in}}}^t(U_{t+1}, V_{t+1})$, the analogue of the objective function in eq. (1) on the set $I_t^{\mathrm{in}}$. By definition $S_{i,j}^* = 0$ for every $(i, j) \in \Lambda_*^c$. Since $I_t^{\mathrm{in}} \subseteq \Lambda_*^c$

$$\tag{73}$$

Note that

$$
\begin{aligned}
U_t V_{t+1}^\top + U_{t+1} V_t^\top - U_t V_t^\top - L^* &= \left(U_t V_{t+1}^\top + U_{t+1} V_t^\top - U_t V_t^\top - U_{t+1} V_{t+1}^\top\right) \\
&\quad + \left(U_{t+1} V_{t+1}^\top - L^*\right) \\
&= (L_{t+1} - L^*) - (U_{t+1} - U_t)(V_{t+1} - V_t)^\top.
\end{aligned}
$$

Inserting this into **??**, by the triangle inequality

$$\mathcal{L}_{I_t^{\mathrm{in}}}^t(U_{t+1}, V_{t+1}) \geq \|L_{t+1} - L^*\|_{F(I_t^{\mathrm{in}})} - \|\Delta U_{t+1}\Delta V_{t+1}^\top\|_{F(I_t^{\mathrm{in}})}.$$

Rearranging the terms yields,

$$\|L_{t+1} - L^*\|_{F(\Omega_t)} \leq \mathcal{L}_{I_t^{\mathrm{in}}}^t(U_{t+1}, V_{t+1}) + \|\Delta U_{t+1}\Delta V_{t+1}^\top\|_{F(I_t^{\mathrm{in}})}.$$

$\square$

*Proof of Lemma D.3.* Our goal is to upper bound $\mathcal{L}_{I_t^{\text{in}}}^t(U_{t+1}, V_{t+1})$. Since $\Omega \setminus \Lambda_t = I_t^{\text{in}} \cup I_t^{\text{out}}$

$$\left[\mathcal{L}_{\Omega_t}^t(U_{t+1}, V_{t+1})\right]^2 = \left[\mathcal{L}_{I_t^{\text{in}}}^t(U_{t+1}, V_{t+1})\right]^2 + \left[\mathcal{L}_{I_t^{\text{out}}}^t(U_{t+1}, V_{t+1})\right]^2.$$

Since $\sqrt{a^2 + b^2} \geq \frac{1}{\sqrt{2}}(a + b)$

$$\mathcal{L}_{\Omega_t}^t(U_{t+1}, V_{t+1}) \geq \frac{1}{\sqrt{2}}[\mathcal{L}_{I_t^{\text{in}}}^t(U_{t+1}, V_{t+1}) + \mathcal{L}_{I_t^{\text{out}}}^t(U_{t+1}, V_{t+1})].$$

Rearranging the terms yields,

$$\mathcal{L}_{I_t^{\text{in}}}^t(U_{t+1}, V_{t+1}) \leq \sqrt{2}\mathcal{L}_{\Omega_t}^t(U_{t+1}, V_{t+1}) - \mathcal{L}_{I_t^{\text{out}}}^t(U_{t+1}, V_{t+1}). \tag{74}$$

Note that $(U_{t+1}, V_{t+1}) = \arg\min\{\mathcal{L}_{\Omega_t}^t(U, V) \mid (U, V) \in \mathcal{B}_\mu \cap \mathcal{C}\left(U_t, V_t, \frac{\delta}{4^{t+1}}\right)\}$. By definition $\mathcal{A}_t \subset \mathcal{B}_\mu \cap \mathcal{C}\left(U_t, V_t, \frac{\delta}{4^{t+1}}\right)$. Therefore for any $(U^*, V^*) \in \mathcal{A}_t$,

$$\mathcal{L}_{\Omega_t}^t(U_{t+1}, V_{t+1}) \leq \mathcal{L}_{\Omega_t}^t(U^*, V^*). \tag{75}$$

Inserting eq. (75) into eq. (74) yields eq. (57) of the lemma. $\qquad\square$

*Proof of Lemma D.4.* By the definition of $\mathcal{L}_{\Omega_t}^t$, see eq. (1), since $X = L^* + S^*$

$$\mathcal{L}_{\Omega_t}^t(U^*, V^*) = \|U_t V^{*\top} + U^* V_t^\top - U_t V_t^\top - L^* - S^*\|_{F(\Omega_t)}.$$

Combining the triangle inequality with the fact that $L^* = U^* V^{*\top}$ gives

$$\begin{aligned}
\mathcal{L}_{\Omega_t}^t(U^*, V^*) &\leq \|U_t V^{*\top} + U^* V_t^\top - U_t V_t^\top - U^* V^{*\top}\|_{F(\Omega_t)} + \|S^*\|_{F(\Omega_t)} \\
&= \|U_t(V^{*\top} - V_t^\top) - U^*(V^{*\top} - V_t^\top)\|_{F(\Omega_t)} + \|S^*\|_{F(\Omega_t)} \\
&= \|\Delta U_t^* \Delta V_t^{*\top}\|_{F(\Omega_t)} + \|S^*\|_{F(\Omega_t)}.
\end{aligned}$$

$\qquad\square$

*Proof of Lemma D.5.* The analogue of the objective function in eq. (1) on the set $I_t^{\text{out}}$ can be written as

$$\begin{aligned}
\mathcal{L}_{I_t^{\text{out}}}^t(U_{t+1}, V_{t+1}) &= \|U_t V_{t+1}^\top + U_{t+1} V_t^\top - U_t V_t^\top - L^* - S^*\|_{F(I_t^{\text{out}})} \\
&= \|\left(U_t V_{t+1}^\top + U_{t+1} V_t^\top - U_t V_t^\top - U_{t+1} V_{t+1}^\top\right) + \left(U_{t+1} V_{t+1}^\top - L^* - S^*\right)\|_{F(I_t^{\text{out}})} \\
&= \|\left(L_{t+1} - L^* - S^*\right) - (U_{t+1} - U_t)(V_{t+1} - V_t)^\top\|_{F(I_t^{\text{out}})}.
\end{aligned}$$

Note that the second term is $\Delta U_t \Delta V_t^\top$. Hence eq. (59) follows by the triangle inequality. $\qquad\square$

### E.3 Proof of Lemma D.8

*Proof.* We denote by $u_i$ and $v_j$ the $i$-th row and the $j$-th column of $\mathcal{P}_\Omega(L - L^*)$, respectively. We denote by $u_i^{(k)}$ the element of $u_i$ with the $k$-th largest magnitude, and a similar definition for $v_j^{(k)}$. As $\Lambda = \text{support}\left(\mathcal{T}_{\gamma\alpha}(L - X, \Omega)\right)$, for any $(i, j) \in \Omega \setminus \Lambda$

$$|(L - X)_{i,j}| \leq \max\left\{\left|(L - X)_{(i,\cdot)}^{(\lceil\gamma\alpha r_i\rceil)}\right|, \left|(L - X)_{(\cdot,j)}^{(\lceil\gamma\alpha c_j\rceil)}\right|\right\}. \tag{76}$$

By assumption 3 $S^* \in \mathcal{S}_\alpha^\Omega$ there are at most $\alpha r_i$ corrupted entries in the $i$th row. Since $\gamma > 1$, the set of $\lceil\gamma\alpha r_i\rceil$ largest entries in the $i$th row of $\mathcal{P}_\Omega(L - X)$ contains at least $\lceil(\gamma - 1)\alpha r_i\rceil$ non corrupted entries. Therefore, the $\lceil\gamma\alpha r_i\rceil$ largest entry in the $i$th row of $\mathcal{P}_\Omega(L - X)$ is smaller than the $\lceil(\gamma - 1)\alpha r_i\rceil$ largest entry in the $i$th row of $\mathcal{P}_\Omega(L - L^*)$. Formally

$$|(L - X)_{(i,\cdot)}^{(\lceil\gamma\alpha r_i\rceil)}| \leq \left|u_i^{\lceil(\gamma-1)\alpha|\Omega_{(i,\cdot)}|\rceil}\right|. \tag{77}$$

The same argument can be applied to each column $j$. Inserting eq. (77) and the analogous inequality for the $j$th column into eq. (76) yields,

$$|(L - X)_{i,j}| \leq \max\left\{\left|u_i^{\lceil(\gamma-1)\alpha \cdot r_i\rceil}\right|, \left|v_j^{\lceil(\gamma-1)\alpha \cdot c_j\rceil}\right|\right\}. \tag{78}$$

Next we upper bound $\left| u_i^{\lceil (\gamma-1)\alpha \cdot r_i \rceil} \right|$. To this end note that,

$$\|u_i\|^2 = \sum_{k=1}^{r_i} (u_i^{(k)})^2 \geq \sum_{k=1}^{\lceil (\gamma-1)\alpha \cdot r_i \rceil} (u_i^{(k)})^2 \geq \sum_{k=1}^{\lceil (\gamma-1)\alpha \cdot r_i \rceil} \left| u_i^{\lceil (\gamma-1)\alpha \cdot r_i \rceil} \right|^2 \geq (\gamma-1)\alpha \cdot r_i \left| u_i^{\lceil (\gamma-1)\alpha \cdot r_i \rceil} \right|^2.$$

Therefore,

$$\left| u_i^{\lceil (\gamma-1)\alpha \cdot r_i \rceil} \right|^2 \leq \frac{\|u_i\|^2}{(\gamma-1)\alpha r_i}. \tag{79}$$

The same argument can be applied to $\left| v_j^{\lceil (\gamma-1)\alpha \cdot c_j \rceil} \right|$. Inserting eq. (79) into eq. (78) gives

$$|(L-X)_{i,j}|^2 \leq \frac{\|u_i\|^2}{(\gamma-1)\alpha r_i} + \frac{\|v_j\|^2}{(\gamma-1)\alpha c_j}.$$

Summing over the entries in $(\Omega \setminus \Lambda) \cap \Lambda_*$ yields,

$$\|L-X\|_{F((\Omega \setminus \Lambda) \cap \Lambda_*)}^2 \leq \frac{1}{(\gamma-1)\alpha} \sum_{(i,j)\in(\Omega \setminus \Lambda)\cap \Lambda_*} \frac{\|u_i\|^2}{r_i} + \frac{\|v_j\|^2}{c_j}.$$

Since $(\Omega \setminus \Lambda) \cap \Lambda_* \subseteq \Lambda_*$,

$$\|L-X\|_{F((\Omega \setminus \Lambda) \cap \Lambda_*)}^2 \leq \frac{1}{(\gamma-1)\alpha} \sum_{(i,j)\in\Lambda_*} \frac{\|u_i\|^2}{r_i} + \frac{\|v_j\|^2}{c_j}$$

$$\leq \frac{1}{(\gamma-1)\alpha} \left[ \sum_{i\in[n_1]} \sum_{j\in\Lambda_{*(i,\cdot)}} \frac{\|u_i\|^2}{r_i} + \sum_{j\in[n_2]} \sum_{i\in\Lambda_{*(\cdot,j)}} \frac{\|v_j\|^2}{c_j} \right].$$

By the assumption $S^* \in \mathcal{S}_\alpha^\Omega$, it follows that $|\Lambda_{*(i,\cdot)}| \leq \alpha r_i$ and $|\Lambda_{*(\cdot,j)}| \leq \alpha c_j$. Therefore,

$$\|L-X\|_{F((\Omega \setminus \Lambda) \cap \Lambda_*)}^2 \leq \frac{1}{(\gamma-1)\alpha} \left[ \alpha \sum_{i\in[n_1]} \|u_i\|^2 + \alpha \sum_{j\in[n_2]} \|v_j\|^2 \right]$$

$$= \frac{2}{\gamma-1} \|\mathcal{P}_\Omega(L-L^*)\|_F^2 = \frac{2}{\gamma-1} \|L-L^*\|_{F(\Omega)}^2.$$

Taking the square root on both sides completes the proof. $\qquad \square$

# F   Proof of Lemma E.1

In our proof we would use the following lemma from Zilber and Nadler (2022) [Lemma SM2.4].

**Lemma F.1.** *Let $L^* \in \mathbb{R}^{n_1 \times n_2}$ be a matrix of rank $r$, and denote $Z^* = \text{b-SVD}_r(L^*)$. Then for any $Z = \binom{U}{V} \in \mathbb{R}^{(n_1+n_2)\times r}$,*

$$d_P^2(Z, Z^*) \leq \frac{1}{(\sqrt{2}-1)\sigma_r^*} \left( \|UV^\top - L^*\|_F^2 + \frac{1}{4}\|U^\top U - V^\top V\|_F^2 \right). \tag{80}$$

*Proof of Lemma E.1.* We prove for $U_t$ and $U_0$. A similar proof follows for $V_t$ and $V_0$. By the triangle inequality

$$\sigma_1(U_{t+1}) \leq \sigma_1(U_t) + \|U_{t+1} - U_t\|_{op}.$$

Hence by induction

$$\sigma_1(U_{t+1}) \leq \sigma_1(U_0) + \sum_{k=0}^{t} \|U_{k+1} - U_k\|_{op} \leq \sigma_1(U_0) + \sum_{k=0}^{t} \|U_{k+1} - U_k\|_F. \tag{81}$$

Since $(U_{k+1}, V_{k+1}) \in \mathcal{C}(U_k, V_k, \frac{\sigma_r^*}{4^{k+1} c_e \sqrt{\kappa}})$, see eq. (8),

$$\|U_{k+1} - U_k\|_F \leq \sqrt{\frac{\sigma_r^*}{4^{k+1} c_e \sqrt{\kappa}}} = \sqrt{\frac{\sigma_r^*}{c_e \sqrt{\kappa}}} \frac{1}{2^{k+1}}.$$

Therefore for $c_e \geq 1$,

$$\sum_{k=0}^{t} \|U_{k+1} - U_k\|_F \leq \sqrt{\frac{\sigma_r^*}{c_e \sqrt{\kappa}}} \sum_{k=0}^{t} \frac{1}{2^{k+1}} \leq \sqrt{\frac{\sigma_r^*}{c_e \sqrt{\kappa}}} \leq \frac{\sqrt{\sigma_r^*}}{\sqrt[4]{\kappa}} \leq \sqrt{\sigma_1^*}. \tag{82}$$

Next we bound $\sigma_1(U_0)$. Denote $Z^* = \begin{pmatrix} U^* \\ V^* \end{pmatrix} = \text{b-SVD}_r(L^*)$. By our assumption $(U_0, V_0) \in \mathcal{B}_{\text{err}}\left(\frac{1}{c_e \sqrt{\kappa}}\right) \cap \mathcal{B}_{\text{bln}}\left(\frac{1}{2c_l}\right)$. Therefore by Lemma F.1

$$d_P^2(Z_0, Z^*) \leq \frac{1}{(\sqrt{2} - 1)} \left(\frac{1}{c_e^2 \kappa} + \frac{1}{16 c_l^2}\right) \sigma_r^* \leq \sigma_1^*. \tag{83}$$

Let $P$ be the minimizer of the Procrustes distance between $U_0$ and $U^*$. Note that $|\sigma_1(U_0) - \sigma_1(U^*)| = |\sigma_1(U_0) - \sigma_1(U^* P)|$. By Weyl's inequality

$$|\sigma_1(U_0) - \sigma_1(U^* P)| \leq \|U_0 - U^* P\|_{op} \leq \|U_0 - U^* P\|_F = d_P(U_0, U^*) \leq d_P(Z_0, Z^*).$$

Hence $|\sigma_1(U_0) - \sigma_1(U^*)| \leq d_P(Z_0, Z^*)$. By the triangle inequality and eq. (83)

$$\sigma_1(U_0) \leq \sigma_1(U^*) + d_P(Z_0, Z^*) \leq 2\sqrt{\sigma_1^*}. \tag{84}$$

Inserting eq. (82) and eq. (84) into eq. (81) completes the proof. $\qquad \square$

