# OpenReview forum: "RGNMR: A Gauss-Newton method for robust matrix completion with theoretical guarantees"
_NeurIPS.cc/2025/Conference — NeurIPS 2025 poster_

### Official Review · Reviewer_hYb1 · 2025-06-23

**Clarity:** 4
**Significance:** 3
**Originality:** 3
**Rating:** 5
**Confidence:** 5

**Summary:**

This paper addresses the problem of matrix completion with corrupted (and unobserved) entries. It proposes to use a Guass-Newton linearization step and outlier removal to improve the assumptions needed for accurate reconstruction. They provide theoretical guarantees for their method and show through experiments that it does well even in challenging settings.

**Questions:**

1) Per the discussion above, could a remark after (2) be added to enhance some intuition?
2) How sensitive is the approach to parameter selection (i.e. number of corruptions, \mu, \alpha...). Remark 3.2 only states this is a common assumption, but are these practical issues or not?
3) Perhaps 2) could be addressed by some sensitivity experiments (dare I say ablation experiments) if they can't be addressed by other works?

**Ethical Concerns:**

["NO or VERY MINOR ethics concerns only"]

**Final Justification:**

After thoroughly reviewing the responses, other reviews, and reviewer discussions, my rating is at an Accept. As discussed in several other places, there are some practical concerns but I believe this to be a nice contribution on the theoretical side.

**Limitations:**

They discussed limitations in the problem approaches.

**Quality:**

3

**Strengths And Weaknesses:**

The problem is well-motivated and the various challenging settings identified. The theoretical guarantees are clear, transparent, and well-explained. I found the entire paper to be very clear and forthcoming.

Figure 1 -- in the caption, it would be helpful to describe what is meant by failure probablity (since the paper studies the robust version, what is failure?)

Remark 2.1 -- what is meant here by runtime savings? It seems that doing additional updates would increase runtime -- is it that the small increase in runtime is far outweighed by the improved convergence (and if so, perhaps this could just be rephrased).

Perhaps a remark after (2) could be added to enhance some intuition -- given that the hope is that this approach allows for a lower number of non-uniformly sampled entries yet also removes outliers, it seems on first glance that it is very possible for the samples to "miss" some directions, making those appear as outliers. In this case, the method actually does itself more damage, but further trying to ignore those directions. Can some intuition be discussed here about this case (rather than waiting for the theory below)?

---

> ### Author Rebuttal · Authors · 2025-07-30
>
> We thank the referee for the positive feedback and constructive comments.
> We first address the concerns raised, and then respond to the specific questions.
> ## Weaknesses
>
> ### Definition of failure:
> To avoid repetitions in the captions of all figures  we defined failure in Line 60 of the manuscript.
> As this seems to have been missed,
> in the revision,
> we will add this to the caption of the figure as well.
>
> ### Remark 2.1:
> We agree with the referee that Remark 2.1 should be made clearer and we will do so in the revision.
> Yes, the small increase in runtime is far outweighted by the improved convergence.
>
> ### Intuition to Equation (2):
> Thanks for the suggestion. We will add an intuitive explanation
> for why Eq. (2) is an effective approach to detect outliers, and does not miss signal directions.
>
> ## Questions
> ### Q1:[Remark after equation 2]
> We will add such a remark in the revision.
>
> ### Q2+Q3: [Parameter selection]
> Our practical algorithm is parameter free and so these are not practical issues.
> A way to select the number of corrupted entries is discussed in Section 2.1 of the manuscript.
> The parameter $\mu$ can also be estimated [1].
>
> [1] R. Sun and Z.-Q. Luo, Guaranteed matrix completion via non-convex factorization, IEEE Trans. Inform.
> Theory, 62 (2016)

---

### Official Review · Reviewer_dKUj · 2025-07-01

**Clarity:** 3
**Significance:** 2
**Originality:** 2
**Rating:** 4
**Confidence:** 3

**Summary:**

This paper introduces a new method for robust matrix completion with strong theoretical guarantees. The algorithm can be described in the following: For robustness, a set of $k$ indices $\Lambda$ is maintained and used to track the estimated outliers based on the largest residual entries. For matrix completion, a factorization method based on a gauss-newton linearization on the factors is used (GNMR by Zilber and Nadler 2022), where only the set of indices $(i,j) \not \in \Lambda$ (or the estimated "inliers") are used to update the factors.

For the outlier indices, $$\Lambda_{t+1} = \arg \underset{\Lambda \subset \Omega, |\Lambda| = k}{\min} \|\|U_t V_{t+1}^T + U_{t+1}V_t^T - U_tV_t^T - X\|\|_{F(\Omega \setminus \Lambda)}^2$$

For the update of the factors,
$$(U_{t+1}, V_{t+1}) = \arg \underset{U,V}{\min} \|\|U_t V^T + U V^T - U_tV_t^T - X\|\|_{F(\Omega \setminus \Lambda_t)}^2$$

Since the number of outliers $k$ must be estimated, the authors also propose a simple yet interesting binary search scheme with justified intuition.

The authors provide recovery guarantees for a slightly modified version of the proposed method. The first theorem establishes the fact that given a sufficiently accurate balanced initialization with bounded row norms, which are relatively standard assumptions, the modified algorithm recovers the true matrix with high probability and linear convergence rate. The recovery guarantees has small sample complexity, and holds the weakest conditions for factorization based methods.

The second theorem establishes the possibility to construct an accurate enough initialization under suitable conditions, such that the recovery guarantee's assumptions can be satisfied. This initialization's success depends on the fraction of outliers to be slightly lower than other algorithms.

The authors also provide a set of simulations comparing against other robust matrix completion methods and demonstrates the effectiveness of the proposed method.

**Questions:**

1. Can the authors provide some insight as to how to choose the rank for the factorization in practice? If possible, can a set of small experiments be added to show the effects of overparametrization? I understand that the time is short, but even a small set of experiments may help.

2. How different are the solutions of RGNMR-BS and the normal variant of RGNMR? Under what conditions may they vary much and how much does it affect the runtime? Are there cases where BS takes a long time to find the correct k and greatly affects the quality of the solution and runtime?

3. Including a set of experiments comparing RGNMR directly with the original GNMR without the robust mechanism may be helpful to understand and validate the effectiveness of the simple robust mechanism.

4. If I recall correctly, GNMR had different variants according to some change of variables. In the experiments for RGNMR, was there a specific variant of GNMR that worked better than the others?

**Ethical Concerns:**

["NO or VERY MINOR ethics concerns only"]

**Final Justification:**

I believe that the paper is a good and interesting in the field of robust matrix completion. The main strengths of the paper is that there is strong theoretical properties of the algorithm proposed, despite the simple extension of the methodology.

My concerns about some details of the practicality, including the estimation of the rank through citing that the method is not sensitive to overparametrization and the option to test different ranks as the input, and initialization through citing that the proposed method works well under random initialization, have been mostly resolved. I am interested to see some of the additional experiments mentioned in the rebuttal in the main paper or the appendix of the paper.

The reviewers all agree that the practicality of the algorithm is still a bit unconvincing, but I believe that the paper has already showed potential through their strong theory and their numerical experiments, and this work should be introduced to the community. I would recommend an accept for the paper to the main conference.

**Limitations:**

yes

**Quality:**

3

**Strengths And Weaknesses:**

Strengths:

The paper provides a simple yet effective algorithm for robust matrix completion, mainly making good use of the existing results from GNMR (Zilber and Nadler 2022). Performance is good compared with other RMC methods with strong theoretical guarantees (especially sample complexity and fraction of outliers for the recovery guarantee), and experiments on a real computer vision dataset is included. The paper is presented in a concise and clear manner.

Weaknesses:

1. I don't see significant novelty in the work, as it mostly relies on the existing work GNMR, which was simple and had strong theoretical properties. The paper's main contribution employs a simple robustness mechanism similar to existing methods to an already powerful matrix completion algorithm.

2. Including a set of experiments comparing directly with the original GNMR without the robust mechanism may help me to understand the true effectiveness of RGNMR. It is unclear to me how effective the robustness mechanism actually is.

3. In applying factorization methods in general, the rank used for the factorization is always difficult to choose. Even for applications where the true rank of a certain matrix under noiseless conditions are known, it may still be difficult, as the best rank to use for matrix completion may be higher than the true rank due to noise and corruption. The authors didn't fully discuss the affect of rank in the factorization and "did not consider the case of overparametrization", and this may limit the practicality of the proposed method.

4. The initialization procedure in theorem 3.2 improves the sample complexity of existing works, but requires a lower fraction of outliers. It is possible that the initialization procedure doesn't provide a sufficiently accurate initialization when data is highly corrupted.

In equation (1) between lines 106 and 107, a square is missing for the norm of the residual.

---

> ### Author Rebuttal · Authors · 2025-07-30
>
> We thank the referee for the constructive feedback and valuable suggestions. We first address the concerns raised, followed by responses to the specific questions.
> ## Weaknesses
> ### No Significant novelty in the work:
> We respectfully disagree with the refeee's assessment that our manuscript has no significant novelty.
> On the methodological side,
> we agree that the proposed method $\texttt{RGNMR}$, can be viewed as a relatively simple combination of outlier removal with the existing scheme $\texttt{GNMR}$.
> However, our manuscript contains at least two significant results: (i) a non-trivial method to estimate
> the number of corrupted entries; and (ii) a theoretical recovery guarantee, which
> improves the state of the art for factorization based methods.
> Deriving these recovery guarantees for $\texttt{RGNMR}$, does not directly follow from the existing guarantees for $\texttt{GNMR}$, which hold only without outliers.
> Furthermore, the recovery guarantees we were able to derive for $\texttt{RGNMR}$
> improve upon those of other RMC factorization based methods in terms of the fraction of corrupted entries. We view this as a significant theoretical contribution.
>
> ### Comparison to $\texttt{GNMR}$:
> $\texttt{GNMR}$ is a non robust method. Like many vanilla matrix completion methods it may fail completely in the presence of even a small fraction of corrupted entries.
> We will illustrate this in the revision.
>
> ### Overparameterization:
> The referee is correct that overparameterization is an important issue when employing factorization based methods.
> For this reason we discuss overparameterization in several places in the manuscript, see for example Figurew 2 as well as Lines 50-67 "Limitations of existing RMC methods".
> Specifically, Fig.2 illustrates
> the performance of $\texttt{RGNMR}$ under overparameterization and show that it  is in fact the only method, out of those tested, that succeeds in this case.
>
> ### Initialization:
> Indeed, the proposed initialization procedure
> yields a highly inaccurate initial guess when the data is highly corrupted.
> However, this does not pose a major problem for $\texttt{RGNMR}$, as empirically $\texttt{RGNMR}$ succeeds even from a random initialization.
>
> ### Additional notes:
> The square is not crucial but we will add it for clarity, thank you for noting that.
>
> ## Questions:
> ### Q1: [Rank estimation]
> First, we note that in various applications in computer vision, the rank of the underlying matrix is known [1].
> That said,
> the referee is correct that in many applications the rank of the underlying matrix is unknown.
> This does not pose a significant challenge for our approach, since
> as illustrated in Figure 2, $\texttt{RGNMR}$
> continues to work well even under overparametrization. For example, for an underlying matrix of rank $r=5$, $\texttt{RGNMR}$ succeeds if its input rank is $r=10$. In short, our method only needs a reasonably tight upper bound on the rank. Another option is to simply run our algorithm with several input ranks. We will discuss this in the revision.
>
>
> [1] Okatani, Takayuki, Takahiro Yoshida, and Koichiro Deguchi. "Efficient algorithm for low-rank matrix factorization with missing components and performance comparison of latest algorithms." 2011 International Conference on Computer Vision. IEEE, 2011.
>
>
> ### Q2:[$\texttt{RGNMR}$ VS $\texttt{RGNMR-BS}$]
> There are indeed settings where accurately
> estimating $k^*$ is difficult. In such cases, $\texttt{RGNMR-BS}$ may fail to recover the matrix, whereas $\texttt{RGNMR}$ (given the exact number of outliers) may still succeed.
> This occurs, for example, when the fraction of corrupted entries is quite large, as illustrated in Fig.5 in the Appendix.
> The difference in runtime between $\texttt{RGNMR}$ and $\texttt{RGNMR}$-BS is illustrated in Fig.10 in the Appendix.
>
> ### Q3: [$\texttt{GNMR}$ VS $\texttt{RGNMR}$]
> We will include such simulations in the revised version.
>
> ### Q4: [$\texttt{GNMR}$ variants]
> In $\texttt{RGNMR}$ we employ an optimization step similar to that of the setting variant of $\texttt{GNMR}$. This variant worked best. We will mention this in the revision.

---

> > ### Comment · Reviewer_dKUj · 2025-08-01
> >
> > Dear authors, thank you for the clarification and the detailed explanation. I have no other questions remaining so far.

---

### Official Review · Reviewer_J6k5 · 2025-07-02

**Clarity:** 2
**Significance:** 3
**Originality:** 3
**Rating:** 4
**Confidence:** 2

**Summary:**

This paper addresses the classic problem of low-rank matrix completion in presence of outliers. It is noted that previous robust matrix completion (RMC) algorithms fail when the number of observed entries in the measurement matrix is relatively small, the low-rank matrices ($U$ and $V$) are overparameterized or the condition number is small. To mitigate these shortcomings, the authors propose an alternation-based algorithm whereby the outlier-corrupted elements are estimated in each step and subsequently excluded from estimating the decomposed matrices.  Since this involves estimating the number of outlier-corrupted entries ($k$) in each iteration, the authors test two versions, i) the oracle case whereby $k$ is known (RGNMR) and ii) the binary search scheme implemented to find an upper bound of $k$.
The authors present a theorem (Theorem 3.1) to provide recovery guarantee under certain assumptions such as small enough fraction of outliers and large enough number of observed entries. The theorem is supplemented by numerical simulation results using largely synthetic data (except for the background extraction example) demonstrating improved performance compared to competitors under various scenarios changing oversampling ratio, fraction of outliers, overparameterized rank, etc.

**Questions:**

1. I wonder if Figures 1 to 3 are drawn assuming some degree of additive noise in each observed entries (like in Fig. 7) or only consider the effect of gross outliers. (In practical viewpoint, assuming no or very little additive noise can be very rare.)

2. I wonder if the authors have tried out various challenging different visibility patterns besides uniform sampling and power-law sampling (Meka et al., 2009) techniques. For instance in structure-from-motion in computer vision (Tomasi and Kanade, IJCV 1992), the visibility pattern of the observation matrix can often be diagonal if the underlying motion of the camera is circular. I wonder if the algorithmic convergence would be severly affected by the ``diagonal-ness'' of the visibility matrix as for other algorithms such as Damped Wiberg/Variable Projection (Okatani et al., Damped Wiberg, ICCV 2011, Hong et al., Revisiting Variable Projection..., CVPR 2017)

3. Since RGMNR is only tested on matrices with comparatively small fraction of outliers (0 to 25%), I wonder if this method would outperform approaches based on non-graduated convexity in which the width of the robust kernel is gradually narrowed in each iteration (e.g., see Le et al., A Graduated Filter Method for Large Scale Robust Estimation, CVPR 2020 --- this is not a paper handling LRMF but a similar technique could be applied).

4. In l. 259, it is reported that RGMNR succeeds even from random initialization. Does this mean all elements of $\Lambda$, $U$ and $V$ are sampled from a particular probability distribution? What is the assumed underlying distribution? I am asking this because from my experience, I feel the convergence would depend significantly on how $\Lambda$ (the predicted outliers) is estimated but is not discussed in the paper.

**Ethical Concerns:**

["NO or VERY MINOR ethics concerns only"]

**Final Justification:**

After considering the paper, reviews and discussions with other reviewers, I am upgrading my score to ``borderline accept'' primarily based on the strength of the theoretical contribution (new recovery guarantee and an estimator for the number of corrupted entries). That said, my practical concerns remain: the non‑oracle variant can lag under simultaneous noise+outliers, experiments are mostly small/synthetic, and runtime/complexity are not compared against fast baselines (e.g. against IRLS-based approaches). During author-reviewer discussion, the rebuttal offered only promises to add larger‑scale experiments, diagonal‑band sampling results, and runtime analyses, but no new evidence or figures were provided, so my confidence in the practical value has not increased.

**For camera‑ready**, including the promises made in the rebuttal, please move the essential experiments into the main paper (simultaneous noise+outliers; non‑uniform/diagonal‑band sampling) and clarify the initialization procedure for the decomposed matrices.

**Limitations:**

Yes

**Paper Formatting Concerns:**

No formatting concern.

**Quality:**

2

**Strengths And Weaknesses:**

[ Strengths ]

- Clarity: the paper is mostly comprehensible despite use of heavy math notations.
- Results: experimental results largely support the claims made in this paper and is mostly superior compared to the competing algorithms.
- Reproducibility: Supplementary code is provided for review.

[ Major weaknesses ]

- Limited motivation: While it is true that low-rank matrix factorization (LRMF) still serves as a baseline model for recommender systems, it is subsequently being more replaced by deep models that are able to capture nonlinear relationships between the user traits and item traits (e.g., Li et al., A Survey on Deep Neural Networks in Collaborative Filtering Recommendation Systems, arxiv). The paper would benefit from a more concrete motivation of why robust low-rank matrix completion is still an important problem (cited works in l.20 are already 4 to 9 years old).

- Novelty of the proposed method: In terms of the algorithmic structure, RGNMR is largely based on the work of GNMR (Zilber and Nadler, 2022) for estimating decomposed matrices. The only major difference is in re-estimating the corrupted entries in each iteration of the algorithm.

- Practicality of the experimental settings: One area where LRMF can still accel compared to deep networks is where the observation matrix ($L$) itself is very large but the number of observed entries is very small (e.g. Bennett et al., The Netflix Prize, Proceedings of the KDD Cup Workshop 2007). Nevertheless, evaluations in this paper is limited to largely synthetic observation matrices of relatively small sizes (3200 x 400). I am not entirely sure if such experimental setting would support the usefulness of this algorithm for real-world applications, e.g. in (large-scale) recommender systems.
Another major weakness in my opinion is that, as demonstrated in Fig. 7, the performance of the (non-oracle) RGNMR-BS is not stronger than others even when the standard deviation of the additive noise is sufficiently small (1e-6 or greater). In practice, additive noise and gross outliers usually occur simultaneously and thus should be the default testing ground in my opinion.

- Runtime and/or algorithmic complexity is an essential component for practical consideration but is not compared against other algorithms.

- Missing connections to other literature: The approach of subsequently improving the set of inliers is similar to the concept of  graduated non-convexity (e.g., Sidhartha et al., Adaptive Annealing for Robust Averaging, ECCV 2024) but is not discussed in the related work.

[ Minor weakness ]

- It seems the positions of figures are slightly unsynchronized compared to the location of paper contents. Perhaps Figures 1 to 3 could be moved to the latter half of the paper? Also, the paper in its current form is not self-contained within the 9 pages limit --- many essential experimental results (e.g. Fig. 7) are in the supplementary material.

---

> ### Author Rebuttal · Authors · 2025-07-30
>
> We thank the referee for their constructive feedback and suggestions. We first address the weaknesses raised,  and then answer the referee's questions.
>
> ## Weaknesses:
>
> ### Motivation:
> We agree with the referee that in recommender systems, matrix completion methods are not state of the art. We will mention this in the revision.
> Despite having been considered for more than 20 years, matrix completion and its robust variant are still important problems in data science.
> Beyond classical applications in computer vision, they are still applicable and used in several contemporary fields, including single-cell analysis in biology **[1]** and sensor networks **[2]**, for example.
> In addition, as noted by numerous recent citations in the manuscript, robust matrix completion is an active field of research in statistics and machine learning.
> We will include a clearer motivation for low rank matrix completion in the revision.
>
> **[1]** Lejun et al, SeqBMC: Single‐cell data processing using iterative block matrix
> completion algorithm based on matrix factorization, IET Systems Biology, 2025.
>
> **[2]** Wu, Di, et al. "Robust low-rank latent feature analysis for spatiotemporal signal recovery." IEEE Transactions on Neural Networks and Learning Systems (2023).
>
>
> ### Novelty:
> We agree with the referee that the proposed method $\texttt{R$\texttt{GNMR}$}$, can be viewed as a relatively simple combination of outlier removal with the existing
> scheme $\texttt{GNMR}$. Two major novelties are: (i) a non-trivial method to estimate the number of corrupted entries; and (ii) a theoretical recovery guarantee, which improves the state of the art for factorization based methods.
> We will clarify these points in the revision.
>
> ### Practicality of the experimental settings:
> In the main text, indeed
> results were presented for matrices of size $3200\times 400$.
> However, please note that
> in Fig. 10 in supplementary of our original submission, we applied $\texttt{RGNMR}$ to  matrices of size up to $16,000 \times 16,000$.
> Namely, $\texttt{RGNMR}$ can be successfully applied to much larger matrices.
> In the revised version, we will add simulations comparing $\texttt{RGNMR}$ to other methods for such large matrices.
>
> ### Additive Noise:
> Fig. 7 compares the reconstruction error of various algorithms in the presence of both outliers and small additive noise. However, the setting here was chosen so that other methods will succeed, specifically
> a small condition number $\kappa=2$,  a high oversampling ratio $\frac{|\Omega|}{r\cdot(n_1+n_2-r)} = 12$, a small fraction of outliers $\alpha = 5\%$ and all methods were given the correct rank.
> Changing one of these parameters, for example increasing the condition number to 10, causes most other methods to have an $O(1)$ error, at any noise level. In contrast,  $\texttt{RGNMR}$ and $\texttt{RGNMR-BS}$ maintain an error proportional to the standard deviation of the noise.
> Therefore, $\texttt{RGNMR}$ is a competitive method even in the presence of additive noise.
> We will add figures to illustrate this in the revised version.
>
> ### Runtime and/or algorithmic complexity:
> Thanks for raising this important issue. Some matrix completion methods, such as those based on gradient descent are very fast.
> In comparison to such methods,
> $\texttt{RGNMR}$ is indeed more computationally demanding.
> Please note that in Figure 10, we did study the runtime of our method as a function of matrix size. As mentioned in the main text P9, L309, the compleixity of $\texttt{RGNMR}$ emprically scales as $O(n^2)$. Moreover,
> as illustrated in Fig.10 its runtime on a standard PC remains reasonable even for large matrices (several minutes for a $16k\times 16k$ sized matrix).
> Improving $\texttt{RGNMR}$ runtime is an interesting problem for future research.
> We will add a discussion of these issues in the revised version.
>
> ### Missing connections to other literature:
> We thank the referee for pointing this out and we will discuss it in revision.
>
> ### Missing essential results from the main text:
>  In revision, with an additional allowed page, we will make the manuscript more self contained, in particular moving some of the figures from supplementary to the main text.
>
> ## Questions
> ### Q1:[Additive noise]
> Indeed, Figures 1 to 3 correspond to simulations with outliers but otherwise no additive noise. In the revision, we will clarify this issue and add figures with a small amount of additive noise (beyond the outliers).
>
> ### Q2: [non-uniform sampling pattern of observed entries]
> Thanks for raising this interesting issue. Following the referee's question,
> we ran $\texttt{RGNMR}$ and several other methods mentioned in our paper, in a setting where the observed entries are concentrated in a diagonal-band pattern as in **[1]**.
> In this setting, all tested methods require a higher number of observed entries to succeed. Yet, $\texttt{RGNMR}$ still significantly outperformed the tested competing methods even in this case.
> In the revision, We will add a comparison between the different methods under this non-uniform sampling pattern.
>
> **[1]** Okatani, Takayuki, Takahiro Yoshida, and Koichiro Deguchi. "Efficient algorithm for low-rank matrix factorization with missing components and performance comparison of latest algorithms." 2011 International Conference on Computer Vision. IEEE, 2011.
>
> ### Q3: [graduated non-convexity approaches]
> One of our initial attempts to solve robust low rank matrix completion was to develop a method that iteratively reweighted the effect of potential outliers. Specifically, we
> replaced the least squares objective to update the factor matrices by an
> IRLS scheme.
> Since this requires employing an IRLS algorithm for every update of $U$ and $V$ it is very computationally demanding.
> In addition, the performance of this approach was not promising. Hence, we developed the method presented in our manuscript, that detects outliers and removes them completely.
>
> ### Q4: [Initialization]
> To randomly initialize  $U_0$ and $V_0$ we generated two random matrices $Z_1\in\mathbb{R}^{n_1\times r}, Z_2\in \mathbb{R}^{n_2\times r}$
> with i.i.d. entries distributed as $N(0,1)$.
> We then set $(U_0, V_0) = (\tilde{U_1},\tilde{U_2})$ where $\tilde{U_i}\tilde{\Sigma_i}\tilde{V_i}$ is the SVD of $Z_i$.
> We set $\Lambda_0$ as the $k$ largest residual entries in $U_0 V_0^\top - X$. We will add these details in the revision.

---

> > ### Comment · Reviewer_J6k5 · 2025-08-04
> >
> > Many thanks to the authors for the rebuttal, and apologies for a slightly late response.
> > I have read through other reviews and rebuttal as well.
> >
> > I have quick follow-up questions regarding the authors' response.
> >
> > 1. In the response regarding Q2, when you mention "all tested methods require a higher number of observed entries to succeed", do you mean all require a ``thicker'' diagonal entries?
> >
> > 2. What is the intuition behind the proposed initializaiton strategy? Does the initialization approach in Q4 still work for differently scaled or even ill-conditioned observation matrices? Or would that require some form of normalization in addition to the procedure proposed in the paper?
> >
> > Looking forward to the response.

---

> > > ### Author Response · Authors · 2025-08-06
> > > **Comment :**
> > >
> > > ### Q1 :
> > > Yes, we do mean that the methods tested require a thicker diagonal band. Specifically, we tested  $n \times n $ matrices of rank $r$ with a bandwidth of $pr$ across different values of $p$. Note that this results in $ \approx(n+n)p \cdot r$ observed entries which means that $p$ is  approximately the oversampling ratio. For example: with $n=500$ and $r =5$ , out method recovers matrices from p = $12$, whereas other methods have a low success rate at that $p$, some of them fail completely. We will provide all details in the revision.
> > >
> > > ### Q2 :
> > > The intuition behind the random initialization follows from the theoretical analysis. Specifically, in *Theorem 3.1* we require that $U_0, V_0$ are approximately balanced. The random initialization procedure described in *Q4* yields two randomly generated factor matrices that are balanced. We want to emphasize that the initialization procedure proposed in the manuscript is not random and provides better results than the random initialization described in *Q4*. We will make it clearer in the revision.
> > >
> > > ### Additional note:
> > > We would like to once again address the referee's remark regarding the motivation for matrix completion. In addition to the contemporary applications mentioned in our original response, matrix completion is also applicable to causal panel data models **[1]**. This is a topic of considerable interest in a broad range of applications.
> > >
> > > **[1]** Choi, Jungjun, and Ming Yuan. "Matrix completion when missing is not at random and its applications in causal panel data models." Journal of the American Statistical Association (2024)

---

> > > > ### Comment · Reviewer_J6k5 · 2025-08-07
> > > >
> > > > Many thanks again for the prompt and detailed response. I have no further question for the authors.
> > > >
> > > > My main perception of the work is that there is still somewhat substantial lack of (illustration of) practical benefits of the proposed method. At the same time, I am not an expert on the theoretical side of this work (as my confidence level shows) so my review can be downweighted in that perspective.
> > > > I have not decided on the final rating but will try and discuss with other reviewers in a few days to consolidate the rating.

---

### Official Review · Reviewer_zeEZ · 2025-07-03

**Clarity:** 3
**Significance:** 3
**Originality:** 3
**Rating:** 5
**Confidence:** 4

**Summary:**

A new method for robust matrix completion, RGNMR, is introduced to overcome the limitations of existing methods. It is empirically shown that RGNMR has significant advantages when only a limited number of entries is observed, and, moreover, that it copes well with both overparameterization of the rank and with ill-conditioned matrices. Besides, it is shown that RGNMR recovers the underlying matrix exactly under suitable assumptions.

**Questions:**

- Is there a connection (formal or intuitive) between the estimation procedure of the corrupted entries in Alg. 1 and Alg. 2?
- Does this scheme for updating $\Lambda_t$ in Alg. 2 also work decently in practice? Or is there a (significant) gap between theory and practice?
- Could you attribute the observed properties of RGNMR to either the estimation procedure of the corrupted entries, or to the Gauss-Newton based method GNMR? What is the contribution of each of these to the overall method?

**Ethical Concerns:**

["NO or VERY MINOR ethics concerns only"]

**Final Justification:**

This paper presents interesting theoretical results for robust matrix completion, provides promising numerical experiments and is well-written. It seems that the main concern of some reviewers is the practicality of the work. While I understand and can even follow some of these concerns, it is my opinion that this is still a strong paper which deserves to be accepted. Finally, I would also like to mention that I appreciated the adequate rebuttal of the authors.

**Limitations:**

yes

**Quality:**

3

**Strengths And Weaknesses:**

Strengths
- The paper is well-written; ideas are clearly presented.
- The numerical results look promising.
- A theoretical analysis reveals that RGNMR recovers the underlying low-rank matrix exactly under suitable assumptions, and improves upon the best-known results for factorization-based methods.

Weaknesses:
- There is a discrepancy between the method that is theoretically analyzed, and the one that is used in the experiments. Although not uncommon in this field, I believe it would strengthen the paper if the authors could explain in more detail the connection between Alg. 2 and Alg. 1.
- The main novelty of RGNMR is the procedure for estimating the number of corrupted entries $k^\*$, which is then combined with the (existing) GNMR scheme. As mentioned in Remark 2.2, various existing RMC methods require an estimate of $k^*$, and therefore they could be equipped with a similar scheme. This raises the important question whether such an approach may yield a performance similar to that of RGNMR.

Typos:
1. Line 28: Remove / rephrase `both’, since it is followed by three adjectives
2. Line 76: RGNMR performs well on real data
3. Line 166: The vertical pipe for “such that” is doubled
4. Line 188: Maybe good to explicitly define the support?
5. Inconsistency: over-removal versus over removal
6. Line 266: The remain open directions
7. Line 472: The subscript init should be upright
8. Line 583: Eq. eq. (14) -> Eq. (14)

---

> ### Author Rebuttal · Authors · 2025-07-30
>
> We thank the referee for their constructive feedback and suggestions. We first answer their three questions and then reply to other issues raised.
>
> **Q1 :** [Connections between Alg.1 and Alg. 2]: In both Alg. 1 and Alg. 2  the estimation procedure of the corrupted entries is based on the magnitude of the residual entries.
> The method employed in Alg. 2 is based on the assumptions made on the corrupted entries, which facilitate deriving theoretical guarantees. Specifically,
> under Assumption 3, the fraction of corrupted entries in each row and column is bounded. Hence, it makes sense to remove only a bounded number of entries from each row and column.
> Accordingly, Alg.2 removes the largest $\gamma \alpha$ residual entries in each row and column.
> However, without any assumptions on the corruption pattern, a more general approach is to remove the largest residual entries across the entire matrix, as employed in Alg.1.
> We will clarify this in the revision.
>
> **Q2 :** [Alg.2 in practice] This is a great question.
> As mentioned above it make sense to use the scheme of Algorith 2 to detect the outlier entries only under Assumption 3.
> Following the referee's question we tested the performance of RGNMR combined with the scheme of Alg. 2.
> Though this method is far less successful than our proposed one it still works decently.
> Specifically, it successfully recovers a $500\times 500$ matrix of rank $5$ and a condition number of $10$, with an oversampling ratio of $12$ and **1%** fraction of corrupted entries.
> This makes it competitive with other RMC methods, which fail to recover such ill-conditioned matrices.
> We will discuss this in the revision.
>
> **Q3:** [ Properties of $\texttt{RGNMR}$]
> The fact that  $\texttt{RGNMR}$ performs well on ill-conditioned matrices, with a small number of observed entries and under overparameterization should be attributed to the optimization step, which updates $U$ and $V$ simultaneously and globally.
> Other optimization schemes, such as gradient based schemes, have been shown to perform poorly under these conditions even when no entries are corrupted.
> However, we would like to clarify that  $\texttt{GNMR}$, as well as many other low-rank matrix completion methods, is not robust to the presence of outliers.
> It may fail miserably even in the presence of very few corrupted entries.
> Therefore, the estimation procedure of the corrupted entries is crucial for the method success.
> We will clarify this in the revision, and also add a figure showing the failure of GNMR in the presence of outliers.
>
> Other issues:
>
> - **Novelty/Estimating number of outliers:**
> We agree that our scheme to estimate the number of outlier entries is an important and novel component of  $\texttt{RGNMR}$. However, accurately estimating the number of outliers is not sufficient.
> Specifically, as mentioned in P9 L297-301 of our manuscript, and illustrated in the simulations, even though we provided $\texttt{AOP}$ and $\texttt{RPCA-GD}$ with the exact number of corrupted entries, they still perform poorly in comparison to  $\texttt{RGNMR}$ and  $\texttt{RGNMR-BS}$.
> We will clarify this in the revision.

---

> > ### Comment · Reviewer_zeEZ · 2025-08-03
> >
> > Dear authors, thank you for your answers. I have no further questions.

---

### Decision · Program_Chairs · 2025-09-17

**Decision:**

Accept (poster)

**Comment:**

The paper suggests a new method for robust matrix completion (which is a combination of the matrix completion and robust PCA problem), building on the recent GNMR algorithm from 2022.  This new method is better at handling outliers, and also takes care of over-estimating the rank.

Reviewers all liked it, especially for having good theory (assumptions are reasonable) as well as good practical performance.  The writing quality was also sufficiently high.

The topic itself is relevant to a good portion of NeurIPS readers

Overall, it seems to be a solid paper and on an interesting topic, hence I'm pleased to recommend acceptance.